# SyntheMol-RL: a flexible reinforcement learning framework for designing easily synthesizable antibiotics

Kyle Swanson [1,4], Gary Liu [2,4], Denise B Catacutan [2,4], Stewart McLellan[2], Autumn Arnold[2], Megan M Tu[2], Eric D Brown [2], James Zou [1,3 ✉] & Jonathan M Stokes [2 ✉]

## Abstract

The rise of antibiotic-resistant pathogens such as *Staphylococcus aureus* has created an urgent need for new antibiotics. Generative artificial intelligence (AI) has shown promise in drug discovery, but existing models often fail to propose compounds that are both effective and synthetically tractable. To address these challenges, we introduce SyntheMol-RL, a reinforcement learning-based generative model that can rapidly design synthetically accessible small-molecule drug candidates from a massive chemical space of 46 billion compounds. SyntheMol-RL improves upon our prior Monte Carlo tree search (MCTS)-based SyntheMol model by generalizing across chemically similar building blocks and enabling multi-parameter optimization. We applied SyntheMol-RL to generate candidate antibiotics against *S. aureus* by optimizing for both antibacterial activity and aqueous solubility, and we found that SyntheMol-RL generated molecules with improved predicted properties compared to both the previous MCTS version of SyntheMol as well as an AI-based virtual screening baseline. We synthesized 79 SyntheMol-RL compounds that were unique relative to the training dataset and found that 13 showed potent in vitro activity, of which seven passed our structural novelty filters that compared them to known antibiotics. Furthermore, one hit compound, synthecin, demonstrated efficacy in a murine wound infection model of methicillin-resistant *S. aureus* (MRSA). These results validate SyntheMol-RL's ability to generate synthetically accessible candidate antibiotics and position SyntheMol-RL as a powerful tool for drug design across therapeutic domains.

**Keywords** Generative AI; Reinforcement Learning; Small Molecule Design; Chemical Synthesizability; Antibiotic Discovery
**Subject Categories** Biotechnology & Synthetic Biology; Computational Biology; Microbiology, Virology & Host Pathogen Interaction

## Introduction

The rapid spread of antibiotic resistance is a critical challenge facing modern medicine, with global implications for public health. In 2019, ~4.95 million deaths were linked to infections caused by drug-resistant bacteria, and this number is expected to rise to 10 million annually by 2050 if the emergence of antimicrobial resistance (AMR) continues to outpace the development of new antibiotics (Murray et al, 2022). Among the most concerning are the ESKAPE pathogens, six bacterial species notorious for their virulence and resistance to multiple drugs (Rice, 2008). Among these, *Staphylococcus aureus* stands out as a substantial cause of morbidity and mortality in both community and healthcare settings (Enright et al, 2002; Miller and Cho, 2011). In high-income regions, *S. aureus* is the leading cause of deaths associated with AMR (25.4% of such deaths), and globally, in 2019, over 700,000 deaths associated with AMR were linked to *S. aureus* infections (Murray et al, 2022). Furthermore, among drug-resistant pathogens, methicillin-resistant *S. aureus* (MRSA) had the greatest increase in attributable burden from 1990 to 2021, doubling from 57,200 to 130,000 attributable deaths over this period. The World Health Organization highlights *S. aureus* as a high priority for the development of new antibiotics, particularly due to the dwindling number of treatment options available for antibiotic-resistant *S. aureus* infections (Tacconelli et al, 2018). Thus, there is an urgent need for novel therapeutic approaches to combat *S. aureus* and its increasing antibiotic resistance.

Artificial intelligence (AI) is playing an important role in drug development broadly and in antibiotic discovery in particular (Stokes et al, 2020; Wong et al, 2024; Wan et al, 2024; Melo et al, 2021). One AI method commonly employed in drug discovery is property prediction models, which are trained to predict the properties of molecules such as antibacterial activity or aqueous solubility (Yang et al, 2019). These models can be applied in a virtual screening (Kitchen et al, 2004) approach to evaluate large chemical libraries to identify compounds that are predicted to be effective. However, since property prediction models must evaluate molecules one-by-one, they do not scale well to the chemical spaces of billions of molecules that are increasingly used in industrial and academic settings. Indeed, in recent years, multiple

[1]Department of Computer Science, Stanford University, Stanford, CA, USA. [2]Department of Biochemistry and Biomedical Sciences, Michael G. DeGroote Institute of Infectious Disease Research, David Braley Centre for Antibiotic Discovery, McMaster University, Hamilton, ON, Canada. [3]Department of Biomedical Data Science, Stanford University, Stanford, CA, USA. [4]These authors contributed equally: Kyle Swanson, Gary Liu, Denise B Catacutan. ✉E-mail: jamesz@stanford.edu; stokesjm@mcmaster.ca

ultra-large synthetically accessible chemical databases have been assembled, including Enamine's REAL Space, WuXi's GalaXi, and OTAVA's CHEMriya, all exceeding $10^{10}$ molecules. Therefore, more recently, generative AI methods have been developed to directly design molecules with promising properties without a slow evaluation process (Bilodeau et al, 2022; Gangwal and Lavecchia, 2024; Bian and Xie, 2021; Zeng et al, 2022).

Generative AI methods for drug discovery are promising, but a major limitation of many of these methods is the synthesizability of the compounds proposed by these models (Gao and Coley, 2020). Although many of the AI-generated molecules are predicted to be effective, they are often synthetically intractable, thus preventing experimental validation and limiting the real-world impact of these models. To address this challenge, we previously developed SyntheMol (Swanson et al, 2024a), a generative AI method for drug design that uses a Monte Carlo tree search (MCTS) guided by a trained antibacterial activity property predictor to explore a space of 30 billion easy-to-synthesize compounds. This method enabled us to rapidly synthesize 58 SyntheMol-generated compounds and experimentally validate their antibacterial efficacy, with six compounds proving to have high potency against *Acinetobacter baumannii* and several other ESKAPE species.

SyntheMol successfully designed compounds with in vitro antibacterial activity, but two important limitations were observed. First, the MCTS algorithm that powers SyntheMol is not the most effective method for exploring chemical space. Indeed, SyntheMol builds compounds by selecting molecular building blocks and combining them with a pre-defined set of chemical reactions to form molecules, and MCTS is responsible for scoring and selecting the most promising building blocks. However, MCTS treats every building block independently and cannot learn patterns of antibacterial activity that are shared across chemically similar building blocks, thereby limiting its search efficiency. Second, SyntheMol can only optimize a single molecular property at a time, even though real-world drug discovery requires identifying compounds with many drug-like properties simultaneously. This proved to be a particular challenge in our prior work, where only two of our six potent antibacterial compounds were sufficiently soluble to be administered for in vivo toxicity experiments.

In this work, we introduce SyntheMol-RL, a significantly enhanced version of SyntheMol that overcomes the two limitations of the original version of SyntheMol (hereafter referred to as SyntheMol-MCTS). First, SyntheMol-RL replaces the MCTS algorithm with a reinforcement learning model that evaluates and selects building blocks in an entirely new way, enabling it to generalize across chemically similar building blocks to more rapidly and effectively search massive combinatorial chemical spaces for promising, easy-to-synthesize compounds. Second, SyntheMol-RL introduces the ability to perform multi-parameter optimization to generate compounds that simultaneously possess multiple relevant molecular properties for drug discovery applications. Moreover, in this work we markedly expand the chemical space SyntheMol-RL explores from 30 billion compounds to 46 billion compounds. Through these advancements, we present substantial improvements in the practicality and efficiency of the original SyntheMol architecture for real-world applications.

We then apply SyntheMol-RL to design easily synthesizable compounds against the target bacterium *S. aureus* with simultaneous optimization for antibacterial activity against *S. aureus* and aqueous solubility (Fig. 1). Importantly, SyntheMol-RL markedly outperforms SyntheMol-MCTS and a virtual screening approach that uses property prediction models in silico, finding more compounds that pass our novelty and diversity filters and are predicted to be antibacterial. Next, we synthesized and experimentally tested 79 compounds designed by two variants of SyntheMol-RL and found two and 11 potent hits, respectively, compared to zero hits for SyntheMol-MCTS and two hits for virtual screening. We further investigated one particularly potent and structurally novel de novo generated compound, which we call synthecin, and we demonstrated its ability to fully arrest the growth of MRSA in a murine wound infection model. These results demonstrate that SyntheMol-RL is an effective and flexible framework for drug design applications.

# Results

## Property prediction model development

SyntheMol-RL is guided by molecular property prediction models that evaluate the properties of generated molecules and provide feedback to improve the generative model's ability to design molecules with desired properties. Therefore, the first step toward building SyntheMol-RL is to train molecular property prediction models on the properties that we require of our molecules. Here, we focus on two key properties: (1) antibacterial activity, which we define as the ability of a compound to effectively inhibit the growth of *S. aureus* in vitro, and (2) aqueous solubility, which is necessary for the compound to be efficiently tested in vivo. For antibacterial activity, we performed our own assay to build a training set, as described below. For aqueous solubility, we took the AqSolDB dataset of 9982 molecules with log solubility values as curated by the Therapeutics Data Commons (Huang et al, 2021). While there are many drug-like properties that we could prioritize at this stage, we chose aqueous solubility to help circumvent issues of low solubility that we observed when we previously applied SyntheMol-MCTS to antibiotic discovery (Swanson et al, 2024a).

To create our antibacterial activity training set, we physically screened an in-house bioactive library of 10,716 diverse compounds against *S. aureus* RN4220 at 50 μM in LB medium (Fig. 2A; Dataset EV1) (Kreiswirth et al, 1983). Experiments were conducted in biological duplicate, with end-point growth being measured at 600 nm optical density ($OD_{600}$) after 16 h of incubation at 37 °C (Fig. EV1A). We computed the mean $\mu$ and standard deviation $\sigma$ $OD_{600}$ value across the library and used $\mu - 2\sigma$ as a threshold for binarizing the $OD_{600}$ values into active (low $OD_{600}$) and inactive (high $OD_{600}$) molecules. After removing duplicate compounds based on SMILES (see "Methods"), we obtained a set of 10,658 unique molecules with 1137 active compounds (10.7%) and 9521 (89.3%) inactive compounds. A t-SNE visualization of our training dataset compounds and 1007 molecules from ChEMBL (Zdrazil et al, 2024) with known antibacterial activity (Dataset EV2) shows that our active compounds cover both known and novel antibacterial chemical space (Fig. 2B).

After acquiring these two training datasets, we leveraged two different property prediction model architectures for each dataset: (1) Chemprop-RDKit and (2) MLP-RDKit (Fig. 2C). The Chemprop-RDKit architecture consists of the graph neural network (GNN) model Chemprop (Yang et al, 2019) augmented with 200

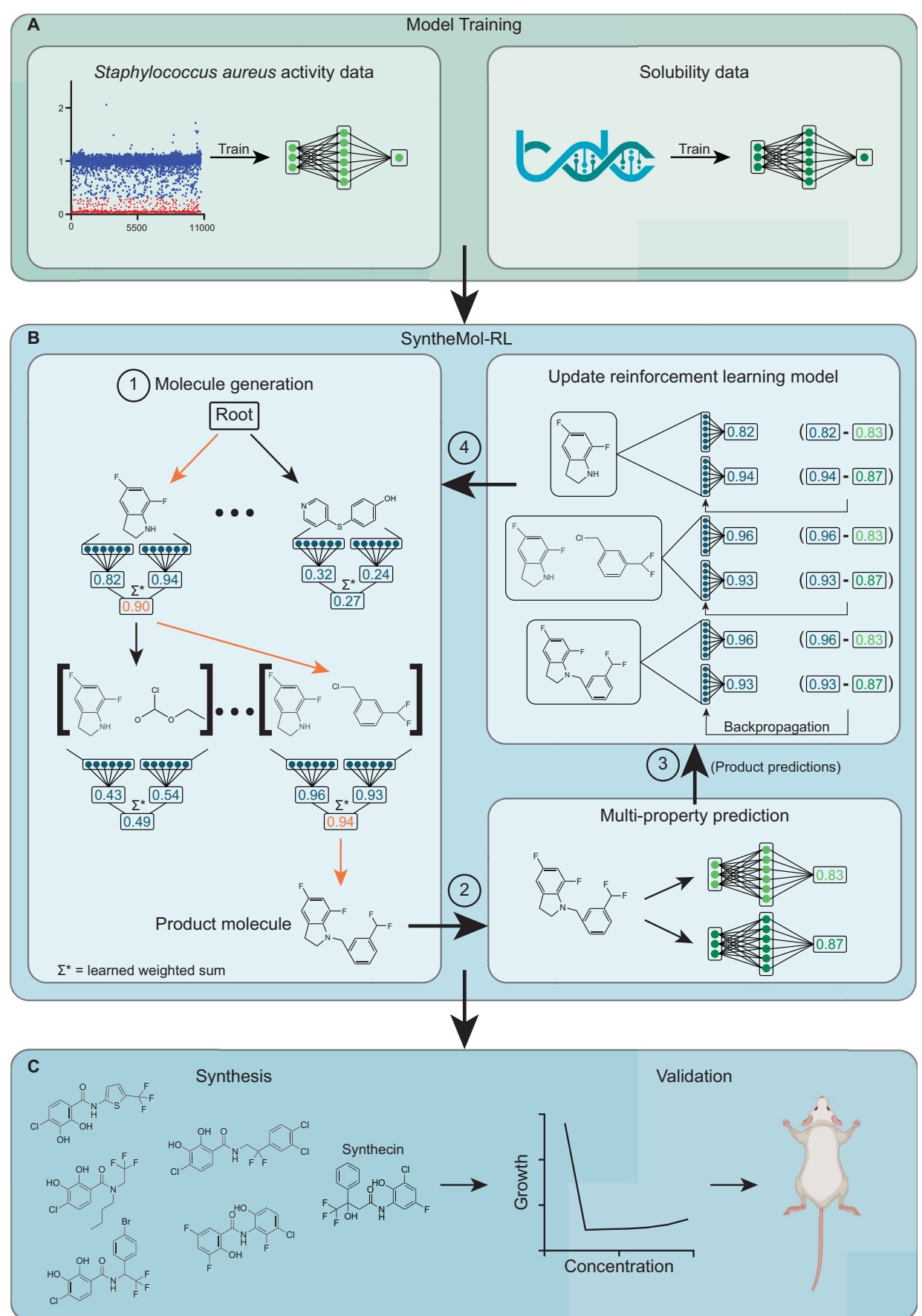

**Figure 1. Overview of the SyntheMol-RL pipeline.**

The SyntheMol-RL pipeline can be broken into three primary steps. (**A**) Property prediction models are trained based on a combination of in-house chemical screening data against *S. aureus* as well as aqueous solubility data obtained from the Therapeutic Data Commons. (**B**) SyntheMol-RL generates molecules guided by these property prediction models. (1) A reinforcement learning (RL) model is used to iteratively select molecular building blocks based on their potential for generating molecules with promising *S. aureus* antibacterial activity and aqueous solubility (RL scores shown in blue). (2) The generated molecule is then scored using the property prediction models, and (3) these scores are used to update the RL model to improve its generative capacity (property prediction scores shown in green). (4) This process is repeated until the desired number of rollouts is reached. (**C**) A prioritized subset of molecules is selected for synthesis and laboratory testing based on predicted properties as well as novelty and diversity. Notably, this SyntheMol-RL pipeline generated a novel compound that can effectively treat a MRSA wound infection in a mouse model, showing the utility of SyntheMol-RL for real-world drug discovery applications.

molecular features computed by the cheminformatics package RDKit (RDKit). Chemprop-RDKit combines the GNN representation of the molecule with the 200 RDKit features using a multilayer perceptron (MLP). The MLP-RDKit architecture is nearly identical to the MLP contained within the Chemprop-RDKit model, but MLP-RDKit only uses the 200 RDKit features as input and does not have a GNN component. We chose to experiment with both model architectures to compare the importance of accuracy and speed; Chemprop-RDKit learns a more comprehensive molecular representation and is thus potentially more accurate, while MLP-RDKit is potentially less accurate but is faster without the GNN component.

We trained Chemprop-RDKit and MLP-RDKit on both the *S. aureus* antibacterial activity and aqueous solubility datasets. For each of the four model-dataset combinations, we trained the model using 10-fold cross-validation with splits containing 80% train, 10% validation, and 10% test data. Each of the four model-dataset combinations trained in less than 80 min on a machine with 8 CPUs and 1 GPU. The Chemprop-RDKit models slightly outperformed the MLP-RDKit models, although both showed high performance (Fig. 2D; Table 1; Dataset EV3), including on more challenging splits based on molecular scaffold (Fig. EV1B,C). The Chemprop-RDKit models were therefore selected for calculating property prediction scores of compounds generated by SyntheMol-RL and by all other generative or screening methods we compared against.

## Generative model development

SyntheMol (Swanson et al, 2024a) is a generative model that designs easily synthesizable molecules with desirable properties, such as inhibition of bacterial growth in vitro. SyntheMol ensures straightforward synthesis of generated molecules by constraining the generative process to a combinatorial chemical space, where it builds molecules using small molecular building blocks that are readily purchasable and chemical reactions that are well-validated. To generate a molecule, SyntheMol selects two or three building blocks and combines them using a chemical reaction to form a molecule, which is then scored by a property prediction model. This score provides feedback to SyntheMol to guide its generative process toward higher-scoring molecules, resulting in a set of generated molecules that are easy to synthesize and likely to possess the desired properties.

In its original form, SyntheMol employed a Monte Carlo tree search (MCTS) algorithm (Kocsis and Szepesvári, 2006; Coulom, 2007) to guide its selection of building blocks and reactions to design molecules. SyntheMol-MCTS treats the generative process as a tree search, where nodes in the tree are combinations of one or more

building blocks. SyntheMol-MCTS computes a value for each node based on statistics about the node that measure both exploration (how many times the node has been selected) and exploitation (the average property score of molecules built using that node). During generation, SyntheMol-MCTS sequentially explores nodes with one, two, or three building blocks, at each step choosing the node with the highest value. When two or three building blocks match a synthesis reaction, a new node is generated with the reaction product. Once a product node is chosen, statistics for all contributing nodes are updated using the molecule's property prediction score, completing one rollout. While MCTS is a simple and effective algorithm, it suffers from a key inefficiency by treating each node independently, rather than leveraging the chemical similarity between molecular building blocks in different nodes to determine the nodes' values.

Here, we developed a new version of SyntheMol that overcomes this important inefficiency by replacing MCTS with a reinforcement learning (RL) algorithm (Zhou et al, 2019; Ståhl et al, 2019; Popova et al, 2018; Olivecrona et al, 2017) (Fig. 1, see "Methods"). SyntheMol-RL uses a deep learning model as a value function that determines the value of a node based on the chemical structures of the building block(s) in that node. Specifically, the RL value function learns to predict the expected property score of molecules constructed using the building block(s) in a given node. During each step of generation, SyntheMol-RL computes the value of nodes containing one, two, or three building blocks. SyntheMol-RL then samples a node proportional to the node's value, which encourages exploration (by not always selecting the highest value node) and exploitation (by sampling higher value nodes more frequently). Importantly, while RL and MCTS can both generate molecules in the same chemical space, the use of RL in place of MCTS represents a major shift in SyntheMol's core generative process, since RL uses a deep learning model to evaluate and select building blocks in a way that is much more flexible, powerful, and generalizable than the statistics-driven approach of MCTS.

To simultaneously optimize for multiple properties (in our case, antibacterial activity and aqueous solubility), the value of a node is computed as a weighted combination of multiple RL models—one for each property. SyntheMol-RL automatically adjusts the weights dynamically over time to explore the portions of chemical space with molecules that exhibit these properties and identify where the greatest density of "hit" molecules exists (those that pass user-defined thresholds for all properties). For the sake of fair comparison, this method of multi-property optimization was also retroactively added to SyntheMol-MCTS. SyntheMol-RL also dynamically adjusts the diversity of generated molecules based on a pre-specified target chemical similarity among generated molecules.

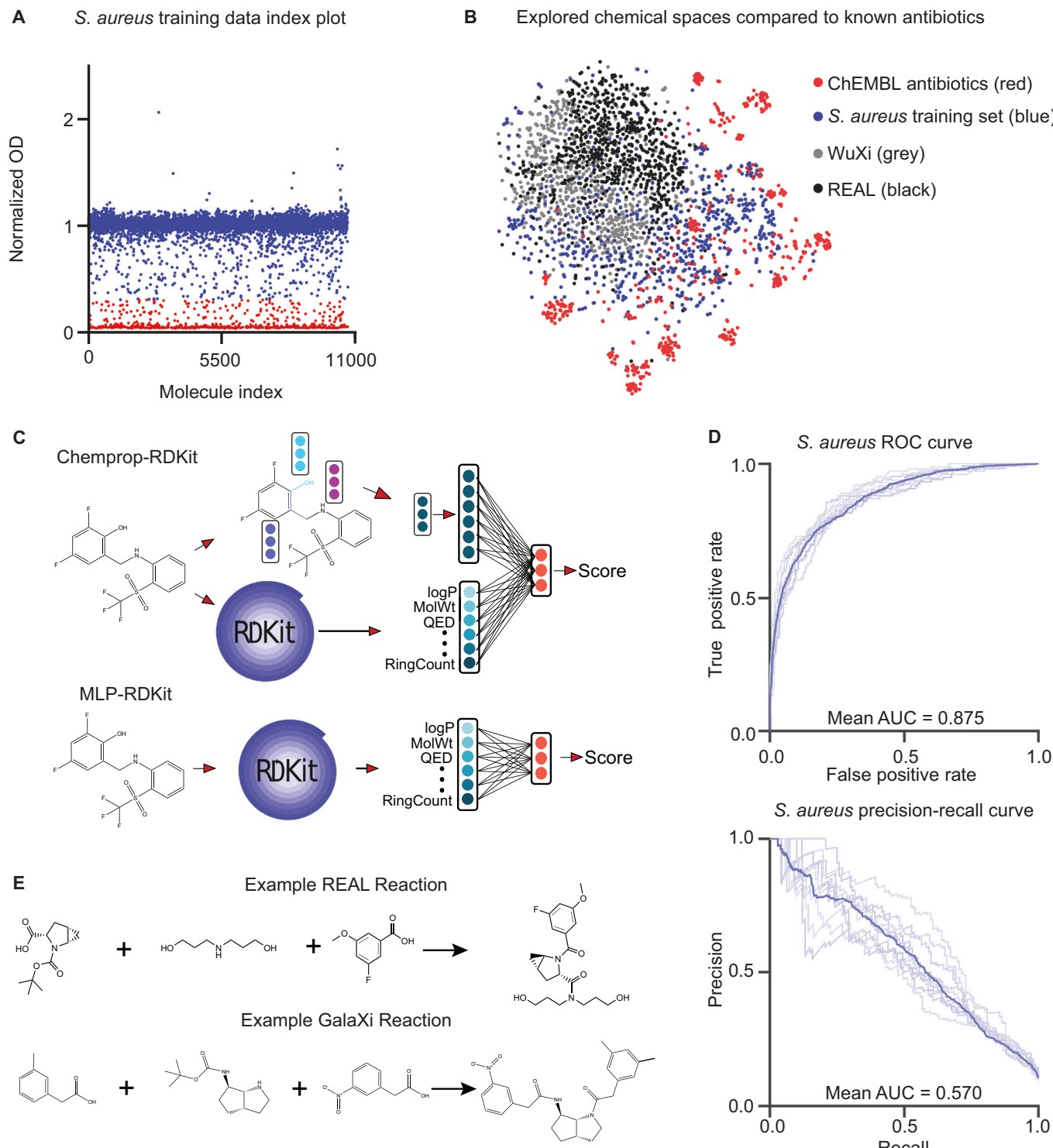

**A** *S. aureus* training data index plot

**B** Explored chemical spaces compared to known antibiotics

- ChEMBL antibiotics (red)
- *S. aureus* training set (blue)
- WuXi (grey)
- REAL (black)

**C** Chemprop-RDKit

MLP-RDKit

**D** *S. aureus* ROC curve

Mean AUC = 0.875

*S. aureus* precision-recall curve

Mean AUC = 0.570

**E** Example REAL Reaction

Example GalaXi Reaction

In SyntheMol-RL, the RL models are trained during the generative process by first collecting generated molecules from previous iterations, along with the path of nodes containing the building blocks that led to that generated molecule. Then, the RL models are trained to predict the property scores of each generated molecule from the nodes that created it. Thus, as the RL model generates more molecules and creates more training data, the accuracy of the RL models and the quality of the generated molecules improve.

For antibiotic generation with SyntheMol-RL, we used our trained Chemprop-RDKit models for antibacterial activity and aqueous solubility as fixed property predictors to score designed

**Figure 2. Property prediction models and chemical spaces.**

(A) A diverse set of small molecules was screened against *S. aureus* RN4220 at 50 μM final concentration. The mean normalized growth across two biological replicates is plotted. The hit cutoff is calculated as: $\mu - 2\sigma$ (the mean minus two standard deviations). Points below the cutoff are considered "hits" and are colored red ($n = 1137$), and points above the cutoff are "non-hits" and are colored blue ($n = 9521$), totaling 10,658 data points used for training. (B) A t-SNE visualization of chemical spaces represented by ChEMBL antibiotics and the *S. aureus* training set in comparison to the chemical spaces explored by SyntheMol-RL. (C) Depictions of the two types of models used in this paper. Chemprop-RDKit (top) was consistently used as the property predictor in each experiment. Both architectures were used for the reinforcement learning model. (D) ROC and precision–recall curves for each model in an ensemble of ten models used as the *S. aureus* activity property prediction model. Dark curves represent the average of all ten models in the ensemble. Area under the curve (AUC) is indicated on the respective plots. (E) Example reactions are shown for the two ultra-large chemical spaces used for SyntheMol-RL: the Enamine REAL Space (top) and the WuXi GalaXi (bottom).

**Table 1. Property prediction model performance.**

| Model | Antibiotic ROC-AUC | Antibiotic PRC-AUC | Solubility MAE | Solubility $R^2$ |
|---|---|---|---|---|
| Chemprop-RDKit | **0.875 ± 0.013** | **0.570 ± 0.050** | **0.656 ± 0.023** | **0.822 ± 0.021** |
| MLP-RDKit | 0.873 ± 0.018 | 0.553 ± 0.040 | 0.688 ± 0.020 | 0.817 ± 0.013 |

Bold values indicate the best performance found across the two models for each metric and task.

molecules since those models were the most accurate (Table 1). For the RL value function, we used either a Chemprop-RDKit model architecture or an MLP-RDKit model architecture to evaluate tradeoffs between accuracy and speed. Briefly, these architectures take molecular graphs or physicochemical property vectors of a node's building blocks as input and then output a prediction of the molecular property (e.g., antibacterial activity or aqueous solubility) for the combination of building blocks in the node. In both cases, we initialized the RL value models with the parameters from our trained property prediction models, but then we further trained them during generation on the RL task of predicting the value of nodes (i.e., combinations of building blocks).

We provided SyntheMol-RL with molecular building blocks and chemical reactions from two easily synthesizable combinatorial chemical spaces: the Enamine REAL Space and the WuXi GalaXi (Figs. 2E and EV1D,E). The Enamine REAL Space (Grygorenko et al, 2020) (2022 version) contains 31 billion molecules that can be produced using ~139,000 molecular building blocks and 169 chemical reactions (Dataset EV4). We used 13 of the most common reactions, which can build 30 billion molecules. The WuXi GalaXi (Chemical Spaces • Ultra-Large Compound Collections by BioSolveIT) (2022 version) contains 16 billion molecules that can be produced with ~15,000 building blocks and 36 chemical reactions (Dataset EV5). Since these two chemical spaces are relatively distinct (Fig. 2B), this gives SyntheMol-RL the capability to generate a structurally diverse set of easily synthesizable molecules within a vast chemical space of 46 billion molecules.

## Generative model application

We applied SyntheMol-RL to generate antibiotic candidates against *S. aureus*. We ran two versions of SyntheMol-RL: (1) RL-Chemprop, which uses a Chemprop-RDKit model architecture as the value function, and (2) RL-MLP, which uses an MLP-RDKit model architecture as the value function. In both cases, we ran the generative model for 10,000 rollouts, resulting in the generation of roughly 10,000 unique molecules. To demonstrate the improved efficacy of RL over MCTS, we also ran MCTS for 10,000 rollouts. Furthermore, to illustrate the benefit of using generative modeling over virtual screening approaches, we applied our Chemprop-

RDKit property prediction models in a virtual screening (Kitchen et al, 2004) manner (VS-Chemprop) to evaluate the antibacterial activity and aqueous solubility of 21 million randomly sampled molecules (14 million REAL and 7 million GalaXi to match the relative sizes of the chemical spaces). We chose 21 million molecules since it took VS-Chemprop about 7 days to evaluate that many compounds, which roughly matched the time of our slowest generative model, RL-Chemprop, on the slowest of our GPUs and CPUs. This gives VS-Chemprop the best possible chance to discover promising compounds compared to the generative models, normalized for time.

We first evaluated the compounds generated by RL-Chemprop, RL-MLP, and MCTS, as well as those screened by VS-Chemprop, based on their predicted antibacterial activity and aqueous solubility. Defining a hit as any molecule with an antibacterial prediction score ≥0.5 and a log solubility ≥ −4 (which constitutes high solubility (Klug et al, 2023)), RL-Chemprop generated 11.6% hits (1273 of 10,983; Dataset EV6) and RL-MLP generated 5.3% hits (493 of 9228; Dataset EV7) compared to 3.0% hits for MCTS (347 of 11,630; Dataset EV8) and 0.006% hits for VS-Chemprop (1290 of 21,000,000; Dataset EV9) (Fig. 3A; Appendix Table S1). The distributions of antibacterial prediction scores (Fig. 3B) and log solubility prediction scores (Fig. 3C) show that all methods can similarly identify soluble molecules, but the generative models— and particularly the RL models—excel at designing molecules with high predicted antibacterial activity. This illustrates the benefit of using RL for molecule design. Of note, we perform a set of ablation experiments that elucidate the effects of different combinations of property-predictor architectures within SyntheMol-RL (Fig. EV2) as well as the effects of individual components of the RL model on generation performance (see "Methods" and Fig. EV3). Importantly, these results demonstrate that optimal values for property weighting and exploration are highly sensitive and inconsistent between RL models, emphasizing the importance of our dynamic weighting mechanism.

To obtain a set of promising compounds for synthesis and testing in vitro, we established a set of four filters to prioritize compounds. First, as above, we identified hits and only kept molecules with an antibacterial prediction score ≥0.5 and a log solubility ≥ −4. Second, we selected novel hits by computing the Tversky similarity (Tversky,

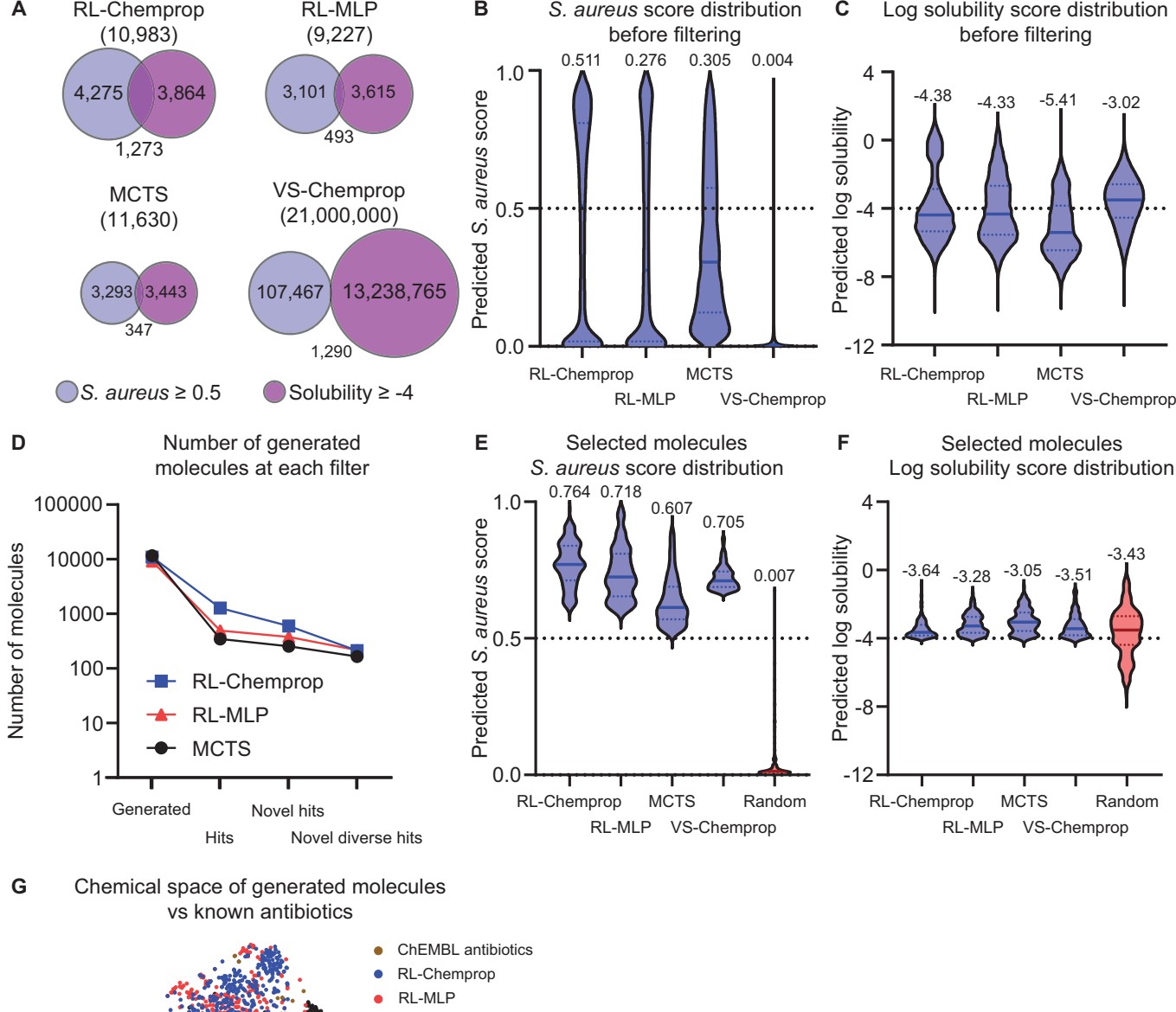

**G**  Chemical space of generated molecules vs known antibiotics

- ChEMBL antibiotics
- RL-Chemprop
- RL-MLP
- MCTS

1977) between each hit molecule and all known antibiotics in our training set and the ChEMBL antibiotics set, only keeping molecules with a maximum Tversky similarity of ≤0.6. Tversky similarity was applied at this point because it prioritizes selecting generated molecules that do not contain a large proportion of the structure of any known antibiotic, independent of the size of the generated molecule. Third, we selected a diverse set of hits by computing the largest set of novel hits such that no two compounds in that set had a Tanimoto similarity (Maggiora et al, 2014) greater than 0.6 (see "Methods"). Fourth, to limit the diverse, novel hits to a practical number of compounds to test in the wet lab, we ranked the remaining compounds by antibacterial prediction score and selected the top 150 compounds from each method. This gave us 600 molecules: 150 each from RL-Chemprop, RL-MLP, MCTS, and VS-Chemprop. Figure EV4

◀ **Figure 3.   Generating compounds with SyntheMol-RL.**

(A) Venn diagrams showing the number of molecules from each SyntheMol method (RL-Chemprop, RL-MLP, MCTS) or virtual screening method (VS-Chemprop) that pass the respective thresholds for predicted *S. aureus* activity and solubility, respectively. (B) Violin plots displaying the distribution of *S. aureus* scores across the four methods (left to right, $n = 10{,}983$; 9228; 11,630; 21,000,000 subsampled to 900,000). Values above each distribution indicate median values. The dotted line indicates the threshold used to define "hit" ($\geq 0.5$). (C) Violin plots displaying the distribution of predicted log solubility scores across four methods. (Left to right. $n = 10{,}983$; 9228; 11,630; 21,000,000 subsampled to 900,000). Values above each distribution indicate median values. The dotted line indicates the threshold used to define "hit" ($\geq -4$). (D) Line graph representing the number of molecules present at each stage of post-hoc filtering for each version of SyntheMol. (E) Violin plots displaying the distribution of *S. aureus* scores across the four methods after applying filters outlined in "Methods" ($n = 150$ each). Random molecules (red) were randomly selected from the chemical space and serve as a baseline for comparison ($n = 150$). Values above each distribution indicate median values. The dotted line indicates the threshold used to define "hit" ($\geq 0.5$). (F) Violin plots displaying the distribution of predicted log solubility scores across the four methods after applying filters outlined in "Methods", as well as random molecules (red) ($n = 150$ each). Values above each distribution indicate median values. The dotted line indicates the threshold used to define "hit" ($\geq -4$). (G) A t-SNE visualization of the training set along with all generated molecules from each of the three SyntheMol models compared to known antibiotics from ChEMBL.

shows a random sample of these molecules from each of the four methods, illustrating their chemical diversity. Finally, to provide an unbiased estimate of the baseline antibacterial activity and solubility of compounds in the REAL and GalaXi chemical spaces, we selected 150 random compounds without any filtering (100 REAL, 50 GalaXi; Dataset EV10), for a total of 750 molecules for in vitro testing (Fig. 3D; Dataset EV11).

Among these 750 selected compounds, the RL-Chemprop and RL-MLP compounds had higher average antibacterial prediction scores (median of 0.764 and 0.718, respectively) compared to MCTS, VS-Chemprop, and random (median of 0.607, 0.705, and 0.007, respectively) (Fig. 3E), while all of the compounds had similar predicted log solubilities (Fig. 3F). This computational evidence shows that SyntheMol-RL is the most adept at generating likely antibiotic candidates.

We then requested the availability of these 750 compounds (747 unique compounds due to three duplicates between RL-MLP and MCTS) from Enamine (659 unique compounds) and WuXi (88 unique compounds), depending on the chemical space each compound belongs to. Among those compounds, 435/659 (66%) of the Enamine compounds and 49/88 (56%) of the WuXi compounds were available for potential synthesis (Dataset EV11; Fig. 3G). Notably, RL-Chemprop, RL-MLP, and MCTS demonstrated a strong preference for Enamine compounds and selected very few WuXi compounds (3, 1, and 0, respectively), of which none were available for synthesis, so WuXi compounds were only available for VS-Chemprop and random. This result may be because the Enamine REAL Space has roughly 10× the number of building blocks as the WuXi GalaXi, despite having only 3× as many total molecules. Since SyntheMol makes building block-level decisions, it has disproportionally more opportunities to select Enamine building blocks.

To further narrow down these available compounds to the most promising set of potential antibiotic candidates, we ranked the available compounds by predicted clinical toxicity according to ADMET-AI (Swanson et al, 2024b; Mukherjee et al, 2025), a multi-task Chemprop-RDKit model for ADMET property prediction. We then selected the 50 compounds with the lowest predicted clinical toxicity for each model type (RL-Chemprop, RL-MLP, MCTS, and VS-Chemprop). For the random baseline, we randomly selected 50 available compounds (33 REAL, 17 GalaXi) from the random set without any filtering.

Altogether, we submitted 250 compounds (50 compounds for each of five settings, 248 unique compounds total with two duplicates between RL-MLP and MCTS) for synthesis

(Dataset EV11). Enamine successfully synthesized 177/224 (79%) unique compounds, while WuXi successfully synthesized 17/24 (71%) unique compounds. Altogether, we obtained 196 unique compounds for in vitro validation (38 RL-Chemprop, 41 RL-MLP, 38 MCTS, 37 VS-Chemprop, and 44 random) (Dataset EV11). Across all stages of filtering and selection, RL-Chemprop performed best on predicted *S. aureus* scores (Fig. EV5) while predicted solubility of compounds remained similar across all four screening strategies (Fig. EV6).

A final computational evaluation was conducted prior to in vitro testing, which benchmarked SyntheMol-RL against GFlowNet (Bengio et al, 2021; Jain et al, 2023) and REINVENT 4 (Loeffler et al, 2024), two state-of-the-art generative models for drug design that are not constrained to the Enamine and WuXi chemical spaces. We modified GFlowNet and REINVENT 4 to optimize for antibacterial prediction score, predicted log solubility, synthetic accessibility (SAScore (Ertl and Schuffenhauer, 2009)), and molecular weight, with the latter two objectives introduced to encourage the generation of compounds that are easy to synthesize. Both models generated molecules that successfully optimized for all four objectives in silico (Fig. EV7; Dataset EV12). However, medicinal chemists at Enamine and WuXi offered limited synthesis of these molecules, with up to 62× higher associated costs and up to 5.7x longer synthesis times relative to the SyntheMol-RL compounds (see Appendix Extended Discussion). This comparison clearly illustrates the advantage of SyntheMol-RL for real-world synthesizable molecule generation over alternative models that rely on imprecise synthesizability heuristics like SAScore.

## In vitro validation of AI-generated molecules

We used growth inhibition assays to determine all synthesized molecules' antibacterial activity against *S. aureus* RN4220, defined using minimum inhibitory concentration (MIC). Remarkably, 11/38 (29%, $P \leq 0.05$ vs all other groups using a Chi-squared test with Bonferroni correction) molecules synthesized from RL-Chemprop and 2/41 (5%) from RL-MLP displayed the ability to completely inhibit bacterial growth at MIC $\leq 8$ µg/ml (Fig. 4A) (Dataset EV11). Of note, MIC $\leq 8$ µg/ml was selected as our activity threshold because this potency represents a reasonable starting point for downstream optimization in antibiotic development campaigns. For comparison, our control set of 44 randomly selected molecules synthesized from both the REAL space and GalaXi was also tested. We observed a complete absence of any activity up to our limit of detection of 128 µg/ml. This may be evidence that these ultra-large

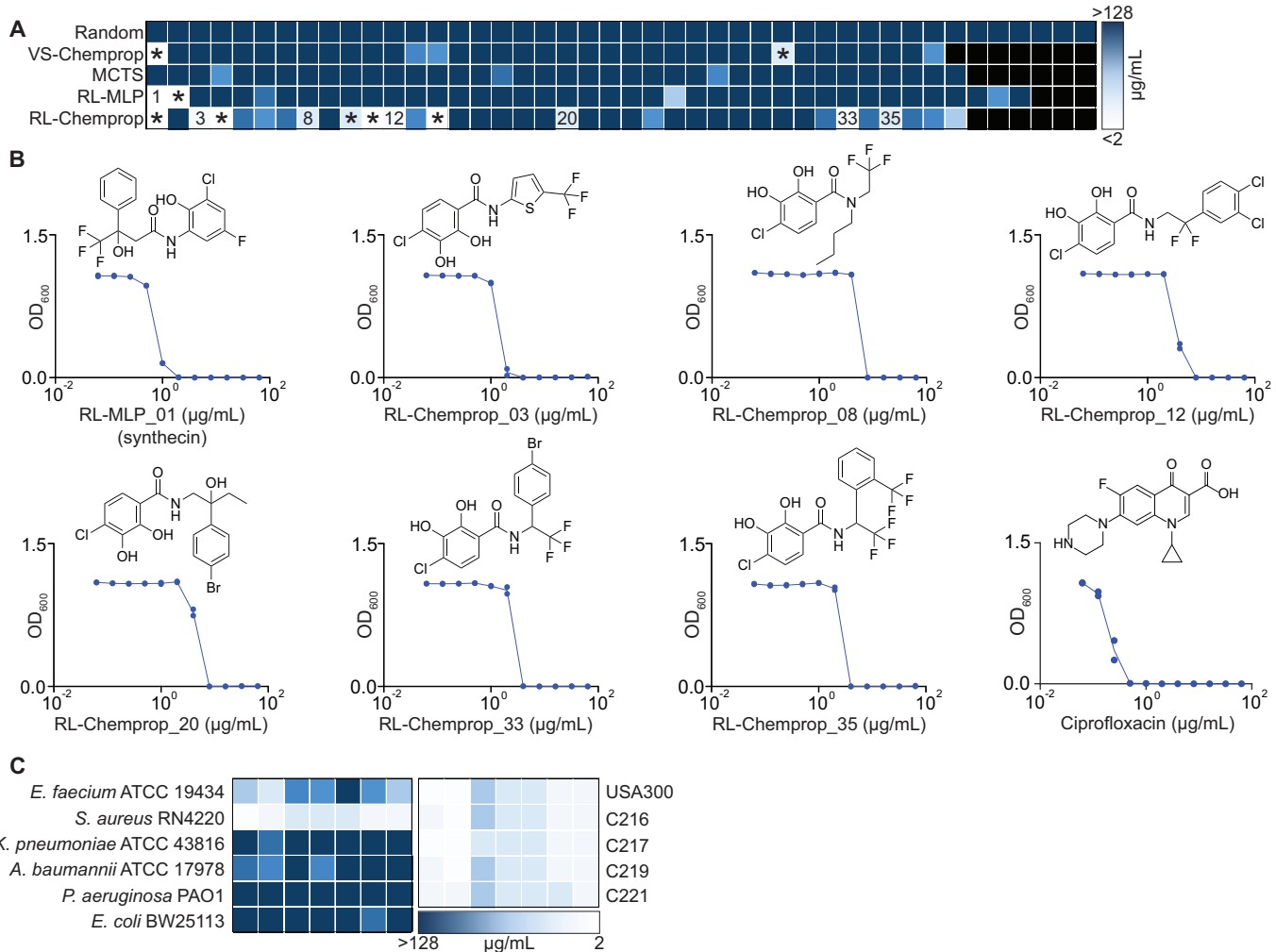

Figure 4. Generated molecules show potent in vitro activity.

(A) A heat map showing the minimum inhibitory concentrations (MICs) of up to 44 compounds from each generative SyntheMol method (RL-Chemprop, RL-MLP, MCTS), the virtual screening method (VS-Chemprop), or the random molecules. Molecules were tested against S. aureus RN4220 in LB medium, with compounds tested at concentrations ranging from 128 µg/ml to 2 µg/ml in twofold serial dilutions. Lighter colors indicate lower MIC values. Experiments were performed in biological duplicates. Annotated squares denote "hit" molecules with MIC ≤ 8 µg/ml. Those denoted with an asterisk (*) indicate molecules that did not pass a thorough manual literature search for structural novelty. (B) Growth inhibition of S. aureus RN4220 by the seven molecules that passed the manual literature search for novelty. Ciprofloxacin is shown as a positive control. Structures of compounds are shown. Experiments were performed using a twofold serial dilution series. Experiments were performed in biological duplicate, with each point representing one replicate. (C) Heatmaps depicting the MICs of the seven compounds against representative isolates of the ESKAPE pathogens (left) and against a representative sample of S. aureus clinical isolates from the CDC AR Isolate Bank (right). Molecules are presented in the same order as b from left to right. Molecules were tested in LB medium, with compound concentrations ranging from 128 µg/ml to 2 µg/ml in twofold serial dilutions. Lighter colors indicate lower MIC values. Experiments were performed in biological duplicates.

chemical databases are non-conducive for antibiotic discovery screens and that common benchmarks from existing physical chemical libraries (~1% hit rate for a standard HTS screen) may not be applicable in this chemical space.

For all compounds deemed a "hit" as defined by MIC ≤ 8 µg/ml, an in-depth manual literature search was conducted to ensure structural novelty from molecules with reported antibacterial activity from the literature. This step is essential to include, as automated filters for novelty employed up to this point remain imperfect. Furthermore, we note that this manual literature search was performed after initial in vitro testing to enable maximally detailed literature analyses on a manageable number of potent molecules. Of the 13 identified hits from

both SyntheMol-RL models, six compounds were found to contain or have structural overlap with salicylanilides (Macielag et al, 1998), a small structural class with known antibacterial activity against Gram-positive bacteria. Specifically, niclosamide (Rajamuthiah et al, 2015; Weiss et al, 2022), a salicylanilide derivative, is currently being investigated for topical treatment of S. aureus and is currently in Phase II clinical trials. A similar literature search was conducted for the two hit molecules sourced from VS-Chemprop; both compounds were found to be similar to molecules with antibacterial activity in the literature.

All molecules that failed the detailed manual novelty check were removed from further investigations, leaving six compounds from RL-Chemprop, one compound from RL-MLP, and no compounds from

VS-Chemprop and MCTS (Fig. 4B). We note that a few of these compounds shared similarity with salicylanilides. However, although synthecin and salicylanilides, like niclosamide, both contain aromatic rings and an amide linkage, these are structurally distinct molecules. Salicylanilide molecules are fairly rigid, defined by a hydroxybenzamide core with strong intramolecular hydrogen bonding between the phenolic hydroxyl and amide carbonyl groups (Suezawa et al, 2000). In contrast, synthecin lacks the salicylanilide core and instead incorporates a conformationally flexible extended aliphatic linker between aromatic moieties, with a tertiary alcohol and trifluoromethyl substitution. Despite synthecin containing both a phenol group and an amide, the phenol is adjacent to the amine end of the amide bond, and therefore cannot participate in the intramolecular hydrogen bonding characteristic of these groups in salicylanilides (Suezawa et al, 2000). Thus, these differences result in distinct molecular geometry and hydrogen bonding, demonstrating that synthecin is a chemically distinct scaffold rather than a salicylanilide derivative. Moreover, while other compounds similar to synthecin have previously been reported with insecticidal or acaricidal activity (Lahm et al, 2021; 李林 et al, 2024), the antibacterial activity of synthecin is a particularly notable observation due to the vast evolutionary distance between bacteria and both insects and arachnids.

For even more certainty, these seven retained molecules were further analyzed for novelty by comparing each compound to the most similar antibacterial compound in the training set and to the most similar ChEMBL antibiotic compound using four similarity metrics: Tanimoto similarity, Tversky similarity, and both the symmetric and asymmetric maximum common substructure ratio (see "Methods"). As seen in Fig. EV8, these seven compounds are structurally distinct from the most similar compounds identified using each of these four metrics.

The remaining seven compounds were assessed for their respective spectrum activity against a phylogenetically diverse panel of ESKAPE isolates (Rice, 2008; Tacconelli et al, 2018), a group of highly virulent and antibiotic-resistant Gram-positive and Gram-negative bacteria that often cause challenging hospital-acquired infections. Molecules displayed no notable potency across all the pathogens (Fig. 4C), including the other Gram-positive pathogen *E. faecium*, indicating all SyntheMol-RL compounds are primarily narrow-spectrum against *S. aureus*.

Importantly, however, the laboratory strain *S. aureus* RN4220 applied until this point in the study may not be representative of the strains commonly responsible for infection. Therefore, we conducted growth inhibition assays against a set of antibiotic-resistant *S. aureus* strains (Lee et al, 2018), especially relevant to the clinic. We screened all seven hit compounds against the most prevalent community-associated MRSA strain for skin and soft tissue infections (SSTIs), USA300, as well as a collection of multidrug-resistant vancomycin-intermediate *S. aureus* (VISA) isolates from the CDC AR Isolate Bank (Lutgring et al, 2018), which covers all resistance mechanisms found within the VISA panel (Fig. 4C). For all seven compounds, no consequential loss of antibacterial potency was found, with all compounds retaining a potent MIC ≤ 8 µg/ml. This provides strong evidence that all seven molecules are able to overcome a wide array of prevalent antibiotic resistance determinants.

## In vivo validation of SyntheMol-RL-generated molecules

*S. aureus* causes most SSTIs in humans, including infected wounds and ulcers, cellulitis, and folliculitis (Lee et al, 2018). Not only is MRSA prevalent in hospitals globally, but community-associated MRSA (CA-MRSA) is responsible for a substantial proportion of SSTIs in Europe (<1 to 32%), Asia (~17%), and the United States (>50%) (Miller and Cho, 2011). SSTIs caused by virulent *S. aureus* (including MRSA) are a risk factor for invasive infections (Lee et al, 2018; Moran et al, 2006), leading to conditions such as bacteremia and endocarditis, which when caused by *S. aureus* are associated with mortality rates of 15% and 30–40%, respectively (Siddiqui and Koirala, 2025). Due to the burden MRSA imposes, we elected to use the most prevalent CA-MRSA isolate in the United States, USA300, as we continued to investigate the translatability of the hit molecules (Miller and Cho, 2011).

Given their potent in vitro activities, we evaluated whether each of the seven hit molecules generated by SyntheMol-RL may be amenable to formulation to treat MRSA-infected wounds. To investigate this possibility, we developed the Chemical Release Evaluation on Agar Media (C.R.E.A.M.) assay, in which a control (10% DMSO) and test molecules (2% w/v) were formulated in Glaxal Base for in vivo application and subsequently applied to MRSA-inoculated LB agar plates, prior to overnight incubation at 37 °C (Fig. 5A). This in vivo mimicking model serves to evaluate the formulation's efficacy as a topical antibacterial treatment. It is worth noting that all seven compounds were readily soluble in the Glaxal Base solution, indicating that the in silico optimization of solubility along with antibacterial activity during generation helped remove potential barriers at this stage in development. We observed that MLP-01 resulted in the largest zone of growth inhibition, indicating that it may demonstrate the most promising in vivo activity in a mouse wound infection model. Interestingly, despite similar growth inhibitory activity in liquid media, there is almost a 2-fold difference in area between MLP-01 and Chemprop-03, showcasing the imperfect correlation between liquid MIC values and growth inhibitory effect when formulated into a cream for in vivo applications. Given its potency and success in the C.R.E.A.M. assay, we continued to characterize the activity of MLP-01, which we named synthecin due to its origin of discovery. Synthecin was tested to determine whether its activity was bactericidal or bacteriostatic against *S. aureus* USA300. We observed bacteriostatic activity in LB medium at 2× and 4× MIC (Fig. 5B). For reference, linezolid, a commonly prescribed antibiotic for the treatment of *S. aureus* skin infections, is also bacteriostatic (Clemett and Markham, 2000).

Based on these encouraging in vitro results, we tested whether synthecin retains its antibacterial efficacy against MRSA USA300 in a mouse wound infection model, a common infection caused by MRSA. To test the in vivo efficacy of synthecin, we established a wound infection in cyclophosphamide pre-treated C57BL/6N mice using *S. aureus* USA300 (~ $2.15 \times 10^7$ CFU) (Fig. 5C). The *S. aureus* infection was allowed to establish for 1 h prior to topical treatments at 1, 4, 8, 12, and 20 h post-infection (hpi) with Glaxal Base supplemented with vehicle (10% DMSO) or synthecin (2% w/v). Mice were then euthanized 24-hpi, and the skin was aseptically dissected and plated to quantify *S. aureus* burden. Vehicle-treated mice carried ~$6.39 \times 10^9$ CFU/g at the experimental endpoint, with wounded tissues exhibiting notable inflammation (Figs. 5D,E and EV9). In contrast, mice treated with synthecin showed a significantly lower bacterial burden, with ~$5.14 \times 10^7$ CFU/g, similar to the bacterial burden observed in pre-treated infection control mice (Appendix Table S2). The synthecin-treated tissues

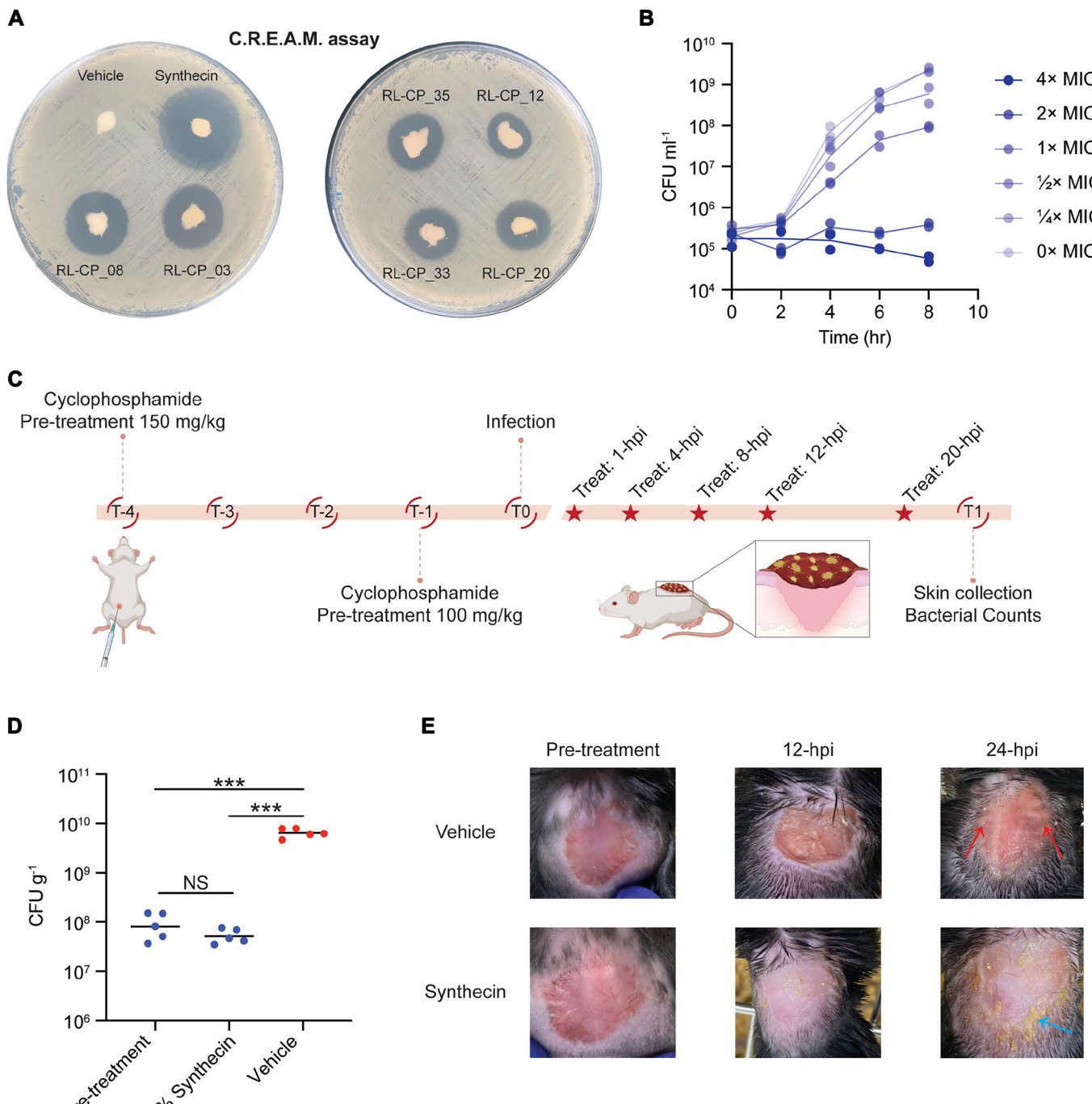

**Figure 5. Synthecin can suppress an *S. aureus* infection in a mouse wound model.**

(**A**) Chemical Release Evaluation on Agar Media (C.R.E.A.M.) assay of vehicle (10% DMSO), MLP-01 (synthecin), RL-CP_08, RL-CP_03, RL-CP_35, RL-CP_12, RL-CP_33, and RL-CP_20 on LB agar. Each molecule was tested at 2% w/v in Glaxal Base. Larger zones of inhibition are expected to correspond with greater efficacy when used as a topical treatment. (**B**) Killing of *S. aureus* USA300 in nutrient-replete conditions (LB) at 4×, 2×, 1×, ½×, ¼×, and 0× MIC for 8 h. Experiments were conducted in biological duplicate. Individual replicates with means connected are plotted. (**C**) Overview of the infection model used to assess the in vivo antibacterial efficacy of synthecin against *S. aureus* USA300. In a wound infection model, mice (C57BL/6N *Mus musculus*) were infected with *S. aureus* USA300 (~ 2.15 × 10⁷ CFU). At 1-h post-infection (hpi), mice were treated with vehicle (10% DMSO) ($n = 5$) or 2% synthecin ($n = 5$). Mice received treatments 4-, 8-, 12-, and 20-hpi. (**D**) Bacterial load of *S. aureus* USA300 from wound tissue prior to treatment ($n = 5$) and of vehicle ($n = 5$) or 2% synthecin-treated ($n = 5$) mice 24-hpi. Pre-treatment represents the bacterial load at the time of initial treatment (1-hpi). Black lines represent the geometric mean of the bacterial load for each group. NS means non-statistically significant. ***$P$ using unpaired two-sided $t$ test with Welch's correction (Pre-treatment vs 2% Synthecin, $P = 0.1774$; Pre-treatment vs Vehicle, $P = 0.0005$; Vehicle vs 2% Synthecin, $P = 0.0004$) (GraphPad Prism version 10.6.0). (**E**) Representative images of the dorsal surface of mice pre-treatment (1-hpi), after 12 h of treatment with vehicle or synthecin, and after 24 h of treatment with vehicle or synthecin. Note the inflammation seen in the vehicle control (red arrows) that is absent in the synthecin-treated mouse. Note the yellow plaques, which is molecule precipitate, seen in the synthecin-treated-mouse (blue arrow) that is absent in the vehicle-treated mouse. Source data are available online for this figure.

also displayed no signs of inflammation (Fig. 5E). Collectively, these data demonstrate that synthecin can effectively suppress an MRSA wound infection, aligning with its exceptional in vitro activity in the C.R.E.A.M. assay.

## Discussion

SyntheMol-RL is an effective generative AI model for designing easily synthesizable small-molecule drug candidates. SyntheMol-RL's use of reinforcement learning enables it to rapidly explore massive combinatorial chemical spaces with tens of billions of molecules for promising compounds that are easily synthesizable by design. Furthermore, the multi-parameter optimization abilities of SyntheMol-RL allow it to identify compounds that simultaneously possess multiple drug-like properties—in our case, antibacterial activity and aqueous solubility—which is a necessity in real-world drug development applications. When applied to antibiotic discovery, SyntheMol-RL generated a superior set of antibiotic candidates for *S. aureus* according to in silico metrics and in vitro experiments relative to both SyntheMol-MCTS (our previous generative model) and VS-Chemprop, a machine learning-based virtual screening method. Indeed, SyntheMol-RL's utility as a flexible tool for readily translatable drug discovery is showcased by the discovery of synthecin, a structurally novel molecule effective at treating a wound infection caused by MRSA in a mouse model.

SyntheMol-RL overcomes limitations present in other generative AI models (Bilodeau et al, 2022; Du et al, 2024) for molecular design. The reinforcement learning component of SyntheMol-RL follows a line of research in reinforcement learning algorithms for molecular generation (Zhou et al, 2019; Olivecrona et al, 2017; Ståhl et al, 2019; Popova et al, 2018), with the sampling of diverse but high-scoring building blocks within the reinforcement learning algorithm inspired by a similar sampling mechanism in GFlowNets (Bengio et al, 2021). However, the SyntheMol-RL algorithm is explicitly designed for synthesizable molecule generation, which requires learning to explore a combinatorial chemical space of tens of billions of molecules constructed from pre-defined molecular building blocks and chemical reactions. Intriguingly, for this reason, in addition to being a generative model, SyntheMol-RL can also be viewed as an intelligent search method, thereby enabling the computationally tractable identification of promising, easily synthesizable molecules from a massive chemical space.

In contrast to SyntheMol-RL, many generative AI models for small-molecule drug design, including reinforcement learning and GFlowNet models, either do not incorporate synthesizability at all (Bengio et al, 2021; Olivecrona et al, 2017) or only include it as a heuristic to be optimized (Jain et al, 2023; Liu et al, 2022), resulting in the generation of many synthetically infeasible compounds, as seen both in our GFlowNet and REINVENT 4 experiments and in other work (Gao and Coley, 2020; Wang et al, 2024; Krishnan et al, 2025). Manual review of AI-generated compounds by medicinal chemists can be used to identify a handful of synthesizable compounds, but it is not scalable to the thousands of compounds generated by these models. While some models have been designed to generate molecules following known synthetic routes (Cretu et al, 2024; Bradshaw et al, 2019; Gottipati et al, 2020; Horwood and Noutahi, 2020; Button et al, 2019; Gao et al, 2022; Klarich et al,

2024; Gao et al, 2025; Koziarski et al, 2024; Seo et al, 2023; Li et al, 2022) like SyntheMol-RL, these studies typically synthesize and experimentally validate few—if any—generated compounds, making it challenging to evaluate those models in terms of real-world synthesizability and biological efficacy. SyntheMol-RL thus represents a major advance beyond these methods by enabling translation of AI-generated compounds into synthesized molecules, thus moving beyond benchmarking in silico metrics to experimental validation in vitro and in vivo.

Interestingly, the molecules generated by SyntheMol-RL tend to form clusters of a particular chemotype. Specifically, in each cluster, SyntheMol-RL selects one shared building block while exploring diverse "second" building blocks (Fig. EV10). This behavior naturally arises from the design of SyntheMol-RL, since the reinforcement learning policy balances both finding building blocks that consistently lead to high-scoring molecules (thus, often repeating a promising building block) while also exploring diverse compounds, which is further reinforced by the post-hoc diversity filtering (thus, selecting diverse second building blocks). The resulting clusters of generated molecules that can be selected for laboratory testing are therefore ideal for both maximizing the probability of finding unique hits across distinct clusters while also naturally exploring the structure-activity relationship (SAR) landscape (Tong et al, 2003) within each cluster. When a cluster of compounds includes multiple hits, as with our RL-Chemprop model, the cluster is likely amenable to SAR optimization based on the presence of those hits and any closely related inactive compounds. In contrast, an isolated hit within a cluster would provide little information about the molecular substructures that are responsible for its activity and may suggest challenges with downstream medicinal chemistry optimization. We note that more diverse (less clustered) compounds could have been generated by reducing SyntheMol-RL's target similarity among generated molecules (set to 0.6 Tanimoto similarity in our experiments), with the potential trade-off of lower property prediction scores; this can be tuned by users depending on their specific applications.

The in vitro results highlight the potential of SyntheMol-RL-generated compounds, particularly in addressing the global burden of MRSA. Indeed, MRSA has seen a dramatic rise in mortality over the past three decades, with deaths associated with MRSA increasing from 261,000 in 1990 to 550,000 in 2021, while attributable deaths doubled to 130,000 in the same period (Murray et al, 2022). In addition, MRSA accounts for 26.1% of attributable deaths, making it the foremost cause of deaths attributed to antimicrobial resistance in high-income regions. To address this urgent and growing unmet need, we designed antibacterial molecules that display promising activity in such resistant *S. aureus* strains. Serendipitously, we discovered molecules with apparent narrow-spectrum activity, which is useful to avoid the widespread disruption of host microbiota and decrease the rate of resistance dissemination (J. Melander et al, 2018; Rea et al, 2011; Jernberg et al, 2007). Among the 79 total SyntheMol-RL compounds synthesized, 13 highly potent active compounds were identified against *S. aureus* RN4220, a remarkable hit rate of 16%. We identified seven compounds that retained potent activity against *S. aureus* USA300, as well as VISA strains that cover all resistance mechanisms present for *S. aureus* in the CDC AR Isolate Bank, further supporting their robust efficacy in targeting burdensome *S. aureus* infections.

Moreover, the C.R.E.A.M. assay provided a simple and informative in vitro platform for evaluating the release and antibacterial activity of molecules formulated for topical administration. The correlation between the zone of inhibition observed in this assay and the subsequent in vivo efficacy of synthecin highlights its utility for prioritizing candidates for in vivo validation. In the mouse wound infection model, synthecin treatment markedly suppressed bacterial proliferation in wounds, resulting in a bacterial burden comparable to pre-treated control mice and mitigated tissue inflammation. These findings demonstrate the C.R.E.A.M. assay as a promising in vitro tool for prioritizing topical antibiotics and the utility of synthecin to effectively treat MRSA-infected wounds.

Building on the successful application to antibiotic discovery, the capabilities of SyntheMol-RL can be further expanded. While SyntheMol-RL succeeded in designing a diverse set of compounds with high property prediction scores that passed our novelty filters, only a subset of the generated compounds showed in vitro activity. This demonstrates the need to improve the accuracy of property prediction models so that high-scoring compounds generated by SyntheMol-RL translate into real hits in the laboratory. The reinforcement learning algorithm within SyntheMol-RL can also be refined for more accurate prediction of which molecular building blocks will produce the most ideal compounds for a given application. The selection of property prediction model architectures within SyntheMol-RL is also an important consideration. Through the ablation experiments (Fig. EV2), we clearly observed the best distribution of *S. aureus* scores when using the more complex Chemprop-RDKit architecture as the RL value model, particularly when the score model architecture was aligned to be Chemprop-RDKit as well (i.e., RL-Chemprop). However, the MLP-RDKit architecture performed slightly better as the RL value model and score model for solubility, so it is worth experimenting with both architectures for new properties. Additionally, while we only ran SyntheMol-RL with two objectives, the model design allows for generating molecules that optimize an arbitrary number of objectives simultaneously. Future work can explore the number and type of objectives that SyntheMol-RL is best suited for to maximize the drug-like characteristics of generated compounds and identify possible limits of the current model when optimizing for challenging combinations of properties.

Notably, since SyntheMol-RL is compatible with any property predictor and combinatorial chemical space, it can be readily extended to a wide variety of drug discovery and molecular design problems. In this work, we prioritized easy and inexpensive synthesis of a relatively diverse set of molecules—as opposed to maximal chemical diversity—in large part due to the practical financial need for highly cost-effective antibiotic design. We therefore selected the Enamine REAL Space and WuXi GalaXi. However, SyntheMol-RL can immediately generate molecules from an exponentially larger chemical space with only a linear increase in runtime by simply changing one setting that allows more than one chemical reaction per generated molecule. The resulting molecules may require custom, multi-step synthesis, and thus will have higher costs and slower synthesis times, but with the benefit of even greater chemical diversity. For even more flexibility, users can substitute the Enamine and WuXi building blocks and reactions for any custom combinatorial chemical space—with building blocks

represented as SMILES and chemical reactions represented as SMARTS—without modifying the SyntheMol-RL algorithm. This gives users maximal freedom to balance ease of synthesis with chemical diversity and novelty, depending on the needs of their drug discovery program. Thus, SyntheMol-RL is a powerful and flexible tool for designing promising compounds that can be rapidly synthesized, thereby bridging the gap between computational design and laboratory validation.

# Methods

**Reagents and tools table**

| Reagent/resource | Reference or source | Identifier or catalog number |
|---|---|---|
| **Experimental models** | | |
| *Staphylococcus aureus* RN4220 | https://www.nature.com/articles/305709a0 Kreiswirth et al, 1983 | RN4220 |
| *Staphylococcus aureus* USA 300 | American Type Culture Collection | BAA-1680 |
| C57BL/6N *Mus musculus* | Charles River Laboratories | C57BL/6 |
| VISA C216 | CDC & FDA Antimicrobial Resistance Isolate Bank; Vancomycin Intermediate *Staphylococcus aureus* (VISA) | SAMN04901606 |
| VISA C217 | CDC & FDA Antimicrobial Resistance Isolate Bank; Vancomycin Intermediate *Staphylococcus aureus* (VISA) | SAMN04901607 |
| VISA C219 | CDC & FDA Antimicrobial Resistance Isolate Bank; Vancomycin Intermediate *Staphylococcus aureus* (VISA) | SAMN04901609 |
| VISA C221 | CDC & FDA Antimicrobial Resistance Isolate Bank; Vancomycin Intermediate *Staphylococcus aureus* (VISA) | SAMN04901611 |
| *Escherichia coli* BW25113 | Coli Genetic Stock Center | CGSC#: 7636 |
| *Pseudomonas aeruginosa* PAO1 | https://www.microbiologyresearch.org/content/journal/micro/10.1099/00221287-13-3-572 Holloway, 1955 | |
| *Klebsiella pneumoniae* ATCC 43816 | American Type Culture Collection | 43816 |
| *Enterococcus faecium* ATCC 19434 | American Type Culture Collection | 19434 |
| *Acinetobacter baumannii* ATCC 17978 | American Type Culture Collection | 17978 |
| **Recombinant DNA** | | |
| N/A | N/A | N/A |
| **Antibodies** | | |
| N/A | N/A | N/A |

| Reagent/ resource | Reference or source | Identifier or catalog number |
|---|---|---|
| **Oligonucleotides and other sequence-based reagents** | | |
| N/A | N/A | N/A |
| **Chemicals, enzymes, and other reagents** | | |
| Small molecules | This study, synthesized by Enamine Ltd and WuXi AppTec | Supplementary Data 11. ['Ordered Molecules'], ['id'] |
| **Software** | | |
| SyntheMol v2.0.0 | This study | |
| GFlowNet v0.1 | https://arxiv.org/abs/2106.04399 Bengio et al, 2021 | |
| REINVENT 4 | https://link.springer.com/article/ 10.1186/s13321-024-00812-5 Loeffler et al, 2024 | |
| **Other** | | |
| Agilent BioTek Synergy Neo2 | Fisher Scientific | BTNEO2 |

## Antibiotic training set curation

The training set consists of the "Bioactives 2" chemical library, an NMR-validated, structurally and functionally diverse library of 10,716 compounds with known bioactivities, including clinical medicines, late-stage molecules in clinical trials, and early-stage molecules undergoing late preclinical optimization (housed in the Centre for Microbial Chemical Biology at McMaster University). The library was screened in two biological replicates against *S. aureus* RN4220 for growth inhibitory activity. Cells were grown overnight at 37 °C in 3 ml Luria–Bertani (LB) medium and then diluted 1/10,000 in fresh LB. In total, 99 μL of cells were added to each well of Costar 96-well flat-bottom plates manually. Each compound was then added (1 μL) to a final screening concentration of 50 μM in a final volume of 100 μL. Plates were then incubated at 37 °C with shaking (900 rpm) for 16 h. Plates were then read at 600 nm using a BioTek Synergy Neo2 plate reader, and data were normalized by plate using interquartile mean (IQM) prior to data compiling and hit identification (see below).

To assign binary activity labels to the screened compounds, we first calculated the average normalized optical density at 600 nm ($OD_{600}$) for each compound using two biological replicates. We then determined the mean and standard deviation of these average normalized $OD_{600}$ values across all compounds in the dataset. A threshold of $\mu - 2\sigma$ was applied to binarize these values—values below this threshold were labeled as active, while those at or above the threshold were labeled as inactive. Subsequently, we canonicalized the SMILES for each compound using RDKit version 2023.9.1. For data points with identical SMILES and binary activity labels, we retained one data point and discarded the others, resulting in 10,660 data points. For data points with identical SMILES but conflicting binary activity labels (at least one labeled as active and one as inactive), we removed all such samples to avoid noise in the activity labels. This process yielded a final *S. aureus* antibiotic

training dataset of 10,658 unique molecules, of which 1137 (10.7%) are active and 9521 (89.3%) are inactive.

## ChEMBL antibiotics curation

To compile a set of known antibiotics for structural comparison, we queried the ChEMBL database on November 8, 2023, using the search terms "antibiotic" and "antibacterial." The search term "antibiotic" yielded 636 molecules (https://www.ebi.ac.uk/chembl/ g/#search_results/compounds/query=antibiotic), with 591 of them having SMILES. The term "antibacterial" returned 611 molecules (https://www.ebi.ac.uk/chembl/g/#search_results/compounds/ query=antibacterial), with 590 having SMILES. We combined the two sets of compounds, excluded molecules with missing SMILES, converted the SMILES to canonical SMILES using RDKit, and then deduplicated the compounds based on the canonical SMILES. This process resulted in a set of 1007 unique molecules.

## Morgan fingerprints

Morgan fingerprints were used for molecular similarity and distance calculations. Morgan fingerprints were calculated with a radius of 2 and 2048 bits using RDKit's GetMorganFingerprintAs-BitVect function.

## Molecular similarities

Molecular similarities were calculated using Tanimoto similarity, Tversky similarity, or maximum common substructure ratio (asymmetric or symmetric).

Tanimoto similarity is defined as $Ta(X, Y) = \frac{|X \cap Y|}{|X \cup Y|}$ where $X$ and $Y$ are Morgan fingerprints of two compounds. It is a symmetric similarity measure with values between 0 and 1, where higher values indicate that compounds are more similar (i.e., they have more molecular substructures in common).

Tversky similarity is calculated as $Tv(X, Y) = \frac{|X \cap Y|}{|Y|}$ where $X$ is the Morgan fingerprint of a proposed (i.e., generated) compound and $Y$ is the Morgan fingerprint of a reference compound. It is an asymmetric similarity measure with values between 0 and 1, where higher values indicate that the proposed compound contains a higher proportion of the reference compound's substructures. We use the Tversky similarity specifically for novelty calculations, where we aim to measure the proportion of known antibiotic functional groups in reference compounds that are contained in proposed compounds while ignoring extraneous substructures in the proposed compounds.

The maximum common substructure (MCS) is the largest contiguous subgraph that is shared between two molecules. The symmetric MCS ratio is defined as $MCS_s(X, Y) = \frac{1}{2}\frac{|MCS|}{|X|} + \frac{1}{2}\frac{|MCS|}{|Y|}$, where $|MCS|$ is the number of atoms in the MCS, $|X|$ is the number of atoms in the proposed (i.e., generated) molecule $X$, and $|Y|$ is the number of atoms in the reference compound $Y$. The asymmetric MCS ratio is defined as $MCS_a(X, Y) = \frac{|MCS|}{|Y|}$, which emphasizes the proportion of the reference molecule that is included in the generated molecule as an exact substructure (similar to the asymmetric nature of the Tversky similarity).

All molecular similarities were calculated with the Python package chemfunc version 1.0.12.

## Molecular diversity

Diverse sets of generated molecules were selected using a method based on the idea of finding a maximal independent set of molecules. The goal of this method is to identify the largest set of molecules such that no two compounds in the set have a similarity (e.g., Tanimoto similarity) greater than some user-defined threshold, which thereby provides a strict guarantee that the selected set of compounds is diverse according to that similarity threshold.

More precisely, given a set of molecules $A$, a molecular similarity metric $s$, and a similarity threshold $t$, we aim to select $D = argmax_{D \subseteq A}\{D | \forall x, y \in D, s(x,y) \leq t\}$. If we reformulate this as a graph $G = (V, E)$ where $V = A$ (i.e., every molecule is a node) and $E = \{x, y \in A | s(x,y) > t\}$ (i.e., an edge connects every pair of molecules with a similarity greater than $t$), then selecting the largest diverse subset of molecules $D$ is precisely equivalent to solving the maximum independent set problem on $G$. A maximum independent set is the largest independent set in a graph, where an independent set is a set of nodes such that no two nodes in the set have an edge connecting them. Since the maximum independent set problem is NP-hard, solving it is intractable, but we can instead calculate a maximal independent set as an approximation (i.e., an independent set that cannot be made larger by adding nodes). Although a maximal independent set is not necessarily the largest independent set in the graph, it is still an independent set, which means it satisfies our diversity requirement of no two molecules exceeding a given similarity threshold.

To obtain a maximal independent set, we formulate a given set of molecules as a graph with nodes and edges as detailed above, and we apply the maximal_independent_set function from networkx version 3.2.1. To slightly improve the approximation, we run the function ten times with different random seeds and take the largest of the maximal independent sets. This set of molecules then serves as our diverse set.

## t-SNE visualizations

t-SNE visualizations were generated using scikit-learn's (Pedregosa et al, 2011) tSNE, applied to the Morgan fingerprints of molecules with Jaccard (Tanimoto) as the distance metric, squared distances, and a principal components analysis (PCA) initialization. For large datasets, a subset of molecules was randomly sampled to accurately represent the dataset in the t-SNE visualization.

## Property predictor architectures

We used two property prediction model architectures: (1) Chemprop-RDKit and (2) MLP-RDKit. Chemprop-RDKit consists of the graph neural network (GNN) Chemprop augmented with molecular features computed by the cheminformatics package RDKit. Chemprop-RDKit takes as input the graph structure of a molecule with atoms as nodes and bonds as edges. Then, the GNN portion of the model applies three message passing steps that aggregate simple features of each atom and bond in a molecule, such as the atom type and bond type, using neural network layers to build vector representations of local neighborhoods of the molecule. After the message passing steps, these local representations are summed to form a single GNN vector representation for the whole molecule. This GNN vector representation, which is 300-dimensional, is then concatenated with a vector of 200 molecular

features computed by RDKit to form a 500-dimensional vector. This combined vector is passed through a multilayer perceptron (MLP) with one hidden layer. The activation function on the output layer of the MLP is a sigmoid layer for binary classification properties (e.g., antibacterial activity) and is a linear layer for regression properties (e.g., aqueous solubility). MLP-RDKit has the same architecture as the MLP at the end of the Chemprop-RDKit model, but it does not have a GNN component and therefore only takes as input the vector of 200 RDKit features. Both architectures were implemented using Chemprop version 1.6.1.

## Model training

Property prediction models were trained using 10-fold cross-validation with data randomly split into 80% train, 10% validation, and 10% test for each fold. The models were all trained on the training data, evaluated on the test data, and used the validation data for early stopping. All models were trained for 30 epochs using the Adam optimizer. The antibacterial activity models used a binary cross-entropy loss and were evaluated using the area under the receiver operating characteristic curve (ROC-AUC) and area under the precision–recall curve (PRC-AUC). The aqueous solubility models used a mean-squared-error loss and were evaluated using mean-squared error (MSE) and the coefficient of determination ($R^2$).

## Enamine REAL space

The Enamine REAL Space comprises 31 billion make-on-demand molecules that can be synthesized through a single chemical reaction using a limited number of molecular building blocks as reactants. For our study, we used the 11/2021 version of the REAL reactions, which includes 169 chemical reactions, along with the 2022 q1-2 version of the REAL building blocks, consisting of 139,517 molecular building blocks. We downloaded the 2022 q1-2 version of the REAL Space on August 30, 2022, which encompasses 31,507,987,117 molecules that can be produced using the specified building blocks and chemical reactions.

Following the approach used in the original SyntheMol-MCTS study, we used only 13 of the most common reactions. To prepare the building blocks for our model, we first used RDKit to convert the SDF file of the building blocks to SMILES format. All building blocks were converted successfully. Next, we deduplicated the molecules by SMILES, reducing the number to 139,444 molecules. We then applied the RDKit salt remover to eliminate salts from the building blocks to avoid incorrect reaction template matching during generation, and we removed 24 molecules whose salts could not be properly removed. This resulted in a set of 137,656 unique molecules (with 139,493 unique building block IDs due to duplicate SMILES). Using the curated reactions and building blocks, we can produce 30,330,025,259 molecules, representing 96.3% of the total REAL Space.

## WuXi GalaXi

The WuXi GalaXi consists of 16 billion make-on-demand molecules. We used the 12/31/2022 version of the WuXi GalaXi, which includes 36 chemical reactions, 15,488 building blocks, and 16,146,071,436 molecules that can be produced using these building

blocks and chemical reactions. To prepare the building blocks for use in our model, we removed building blocks with missing IDs (4 building blocks) and missing SMILES (4 building blocks) and then deduplicated by SMILES, leaving 14,977 unique molecules. We then removed salts from the building blocks using the RDKit salt remover (all salts were correctly removed).

## SyntheMol-MCTS

The original implementation of SyntheMol (Swanson et al, 2024a) used a Monte Carlo tree search (MCTS) algorithm to generate molecules. Briefly, the MCTS algorithm works as follows. The input to the algorithm is a chemical synthesis tree $T$ that consists of a set of nodes $N$ in $T$. Each node $N$ contains $N_{mol}$, which is a set of one or more molecular building blocks $B$ in a chemical space $C$ ($B \subset C$). MCTS defines a value function on nodes, $S(N) = \frac{Q(N) + P(N)U(N)}{D(N)}$. This value function balances exploiting nodes that lead to high-scoring molecules via $Q(N)$, selecting nodes containing building blocks with high property prediction scores via $P(N)$, exploring rarely visited nodes via $U(N)$, and selecting diverse building blocks via $D(N)$ (additional details are available in ref. (Swanson et al, 2024a)).

To generate a molecule, MCTS first computes the value $S(N)$ of every node containing a single building block. The MCTS policy is then to select the node with the highest value. Given this choice, MCTS creates every possible node with this first building block and a second building block that is synthetically compatible with the first building block in some chemical reaction $r$ contained in the set of chemical reactions $R$ defined by the chemical space ($r \in R$). MCTS then scores each of these nodes with two building blocks and applies its policy by selecting the node with the highest value. Next, MCTS creates every possible node with the two selected building blocks and a third synthetically compatible building block, and it also creates nodes for every possible molecule that can be formed by applying one of the chemical reactions in $R$ to just the two selected building blocks to form a new molecule. MCTS scores all of these nodes— both nodes with three building blocks and nodes with a single molecule formed from the two building blocks—and again follows its policy by choosing the node with the highest score. If that node contains three building blocks instead of a single molecule, then nodes are created using all compatible chemical reactions to combine those three building blocks, and MCTS selects the node with the highest value.

At this point, MCTS has completed a single rollout and has obtained a single molecule composed of two or three building blocks. This molecule $m \in C$ is scored by a weighted combination of $L$ property predictors, $M_k : C \rightarrow R$ for $k \in \{1, ..., L\}$, using property weights $w_k$ for $k \in \{1, ..., L\}$ (where $\sum_{k=1}^{L} w_k = 1$ and $0 \leq w_k \leq 1$) to obtain the molecule's overall property score, $p(m) = \sum_{k=1}^{L} w_k * M_k(m)$. This overall property score is then used to update the $Q(N)$ exploit score of every node $N$ selected during the generation of molecule $m$. In addition, the exploration and diversity scores, $U(N)$ and $D(N)$, are updated to reflect the nodes visited during the generation of molecule $m$. MCTS repeats this process for a fixed number of rollouts.

## SyntheMol-RL

The reinforcement learning (RL) version of SyntheMol employs the same synthesis tree $T$, chemical space $C$, chemical reactions $R$, and property predictors $M_1, ..., M_L$ with their weights $w_1, ..., w_L$ as

MCTS, but it reformulates the method for computing node values and selecting nodes during each rollout. In SyntheMol-RL, RL replaces the MCTS value function $S(N)$ with an RL value function $V(N)$. $V(N)$ is implemented as a deep neural network that takes as input the building blocks in the node, $N_{mol}$, and outputs a prediction of the expected overall property score of molecules that can be created from the building blocks $N_{mol}$ by following the RL policy. The RL policy is to apply the RL value function $V(N)$ to all nodes created at a given step in generation and then sample a node proportional to the values of the nodes with a temperature scaling, i.e., $P(N) \propto e^{V(N)/\tau}$. The temperature parameter $\tau$ can be tuned to affect the amount of exploration or exploitation performed by the RL policy; high temperature prefers exploration (more uniform probabilities) while low temperature prefers exploitation (more spiky probabilities).

The RL value function $V(N)$ is implemented as a weighted combination of models, similar to the molecule property score $p(m)$. Specifically, $V(N) = \sum_{k=1}^{L} w_k * Z_k(N_{mol})$ where $Z_1, ..., Z_L$ are deep learning models and $w_1, ..., w_L$ are the same property weights used in $p(m)$. They are then trained on the RL value function objective as follows. After each rollout constructs a molecule $m$, the RL algorithm stores tuples of $(N, M_1(m), ..., M_L(m))$ for every node $N$ in the trajectory of nodes that were selected to create the molecule $m$. This builds an RL value function training set of nodes along with the property prediction scores of the final molecules created from those nodes. After every $n_{rl\ train}$ rollouts, the RL models $Z_1, ..., Z_L$ are trained for $n_{rl\ epochs}$ epochs to take as input the building blocks of the node, $N_{mol}$, and predict the relevant score of the generated molecule from $M_1, ..., M_L$ using a mean-squared-error loss.

## Dynamic RL parameters

The RL temperature parameter $\tau$ and property weights $w_1, ..., w_L$ have a large impact on the diversity of generated molecules and their relative property scores, respectively. By default, the temperature is set to $\tau = 0.1$, and the property weights are set to $w_1, ..., w_L = \frac{1}{L}$. However, these values may not be optimal for every molecule design problem. Rather than manually tuning each parameter, we designed a dynamic tuning mechanism (Hong et al, 2018) that automatically adjusts the temperature and property weights based on pre-defined goals.

The RL temperature aims to balance exploration and exploitation, so we defined the temperature goal explicitly by setting a target similarity $\lambda^*$, which is the desired maximum Tanimoto similarity between each generated molecule and all previously generated molecules. We then defined a dynamic tuning method to adjust the RL temperature during generation to obtain a molecule similarity of $\lambda^*$ on average. During generation, the dynamic temperature tuning mechanism maintains a rolling average molecule similarity $\lambda_{avg}$, which is initialized as $\lambda_{avg} = \lambda^*$. After each rollout $i$ generates a molecule $m_i$, $\lambda_{avg}$ is updated as $\lambda_{avg} = \gamma * \lambda_{avg} + (1 - \gamma) * \max_{j=1,...,i-1} sim(m_i, m_j)$ where $\gamma = 0.98$ is the rolling average weight and $sim(m_i, m_j)$ is the Tanimoto similarity between the molecule $m_i$ generated on rollout $i$ and the molecule $m_j$ generated on rollout $j$. Next, the percent difference between the average similarity $\lambda_{avg}$ and the target similarity $\lambda^*$ is computed as $\lambda_{diff} = \frac{(\lambda_{avg} - \lambda^*)}{\lambda^*}$ and a new desired temperature is computed as

$\tau_{new} = \tau + \lambda_{diff} * \tau$. To smooth out changes to the temperature, the temperature is updated via a rolling average with $\tau = \gamma * \tau + (1 - \gamma) * \tau_{new}$. Finally, the temperature is clipped within reasonable limits with $\tau = \max(\tau_{\min}, \min(\tau, \tau_{\max}))$ where $\tau_{\min} = 0.001$ and $\tau_{\max} = 10$.

The property weights aim to balance the importance of the two property objectives (antibacterial activity and aqueous solubility), so we defined the property weight goal as maximizing the number of molecules that simultaneously possess both properties. Specifically, we defined success thresholds $t_1, ..., t_L$ for each property, and we considered that molecule $m$ is a hit for property $k$ if $M_k(m) \geq t_k$. In a similar manner to dynamic temperature tuning, we perform dynamic property weight tuning by computing a rolling average. Here, for each property $k \in \{1, ..., L\}$, we define a rolling average success rate $s_{avg}^k$ which is initialized with $s_{avg}^k = 0$. Our aim is to maximize all success rates simultaneously, which we hypothesize will occur when the success rates are equal. Therefore, we aim to adjust the property weights to make the success rates match. After each rollout $i$ generates a molecule $m_i$, then $s_{avg}^k$ is updated as $s_{avg}^k = \gamma * s_{avg}^k + (1 - \gamma) * I[M_k(m_i) \geq t_k]$ where $\gamma = 0.98$ as with the dynamic temperature and $I$ is in the indicator function that is 1 if $M_k(m_i) \geq t_k$ and 0 otherwise. We then compute the average success rate across the properties: $s_{avg} = \frac{1}{L} \sum_{k=1}^{L} s_{avg}^k$. If $s_{avg} = 0$, meaning no successful molecules have been generated, then the property weights are unchanged. If $s_{avg} > 0$, then we determine the relative amount by which each individual success rate, $s_{avg}^k$, deviates from their mean, $s_{avg}$, by computing $s_{avg\ diff}^k = \frac{(s_{avg}^k - s_{avg})}{s_{avg}}$. We then perform a rolling average update of the property weights with $w_k = \gamma * w_k + (1 - \gamma) * (w_k - s_{avg\ diff}^k * w_k)$. These property weights are normalized to sum to 1 by computing $w_k = \frac{w_k}{\sum_{k=1}^{L} w_k}$. Then, the property values are clipped to $w_k = \max(w_k, w_{\min})$ where $w_{\min} = 0.001$ and are then renormalized with $w_k = \frac{w_k}{\sum_{k=1}^{L} w_k}$. At a high level, this ensures that if the success rate for property $k$ is higher (or lower) than the average success rate across the properties, then the corresponding property weight $w_k$ will be decreased (or increased) proportional to the difference between that property's success rate and the average success rate.

## Generating molecules with SyntheMol

We applied SyntheMol, both using MCTS and using RL, to generate antibacterial molecules against *S. aureus*. For both RL and MCTS, we had $k = 2$ properties: antibacterial activity (binary classification) and aqueous solubility (regression). Since Chemprop-RDKit performed best on both properties, we set the property predictors $M_1$ and $M_2$ to be the trained Chemprop-RDKit models for antibacterial activity and aqueous solubility, respectively. Specifically, $M_1$ and $M_2$ were the ensemble of ten models for each property from the ten folds of cross-validation, and predictions consist of the average prediction across the ten models. We adapted MCTS to use the same dynamic property weighting scheme as RL to set the weights $w_1$ and $w_2$, and for both models, we defined the success thresholds as $t_1 = 0.5$ for the antibacterial model and $t_2 =$

$-4$ for the log solubility model to match our hit definitions for filtering generated compounds.

For RL, we implemented the RL value function models $Z_1$ and $Z_2$ as either Chemprop-RDKit or MLP-RDKit models. When $Z_1$ and $Z_2$ are Chemprop-RDKit models, we refer to the overall architecture as "RL-Chemprop", and when $Z_1$ and $Z_2$ are MLP-RDKit models, we refer to the overall architecture as "RL-MLP". The $Z_1$ and $Z_2$ models were initialized with the weights of $M_1$ and $M_2$. Specifically, we set $Z_1$ and $Z_2$ to be the first of the ten models in the ensembles of $M_1$ and $M_2$ since $Z_1$ and $Z_2$ in our implementation are single models, not ensembles.

Notably, as an RL value function, these models must make predictions not just on single molecules but on combinations of molecules (e.g., multiple molecular building blocks). For Chemprop-RDKit models, the model itself is unmodified. The Chemprop GNN component is applied in its usual form to multiple molecules by treating them as a single graph with disconnected components, while the RDKit component uses the average of the RDKit features of the individual molecules, thereby preserving the dimensionality of the RDKit feature input. For MLP-RDKit models, the model is modified by replicating the MLP weights of the first layer $n_{\max\ bbs}$ times, where $n_{\max\ bbs}$ is the maximum number of building blocks that can be combined into a single molecule ($n_{\max\ bbs} = 3$). For a given combination of molecules, RDKit features are computed, concatenated, and fed as input to this modified MLP. If there are fewer than $n_{\max\ bbs}$ molecules, the remaining elements of the input vector are set to 0.

After modification as described above, the Chemprop-RDKit ($Z_1$) and MLP-RDKit ($Z_2$) RL value function models were then trained during generation every $n_{rl\ train} = 10$ rollouts for $n_{rl\ epochs} = 5$ epochs across all the nodes and molecules that had been previously generated. To dynamically tune the RL temperature $\tau$, we set the target similarity $\lambda^* = 0.6$ to match our diversity filter, which uses 0.6 as a threshold to define diverse molecules.

We ran all three SyntheMol versions—RL-Chemprop, RL-MLP, and MCTS—for 10,000 rollouts, generating 10,983; 9228; and 11,630 molecules, respectively. We used 8 CPUs for all three versions, and we additionally used 1 GPU for RL-Chemprop due to the computational burden of running the Chemprop models on millions of building block combinations (RL-MLP and MCTS were faster with CPU only due to the relatively lightweight computations involved). RL-Chemprop took 4–7 days, RL-MLP took 9–12 h, and MCTS took 4–5 h, depending on the precise hardware that was used (GPUs were either NVIDIA A40, TITAN_Xp, TITAN_V, or RTX_2080Ti).

## Virtual screening with Chemprop-RDKit

To perform virtual screening, we first selected 21 million molecules by sampling 14 million molecules uniformly at random from the 31 billion REAL molecules and by sampling 7 million molecules uniformly at random from the 16 billion GalaXi molecules. The final 21 million compounds roughly approximated the ratio between the two chemical spaces. We then applied our Chemprop-RDKit models for antibacterial activity and aqueous solubility to all 21 million molecules using 1 GPU and 8 CPUs, which took roughly 7 days. This is in line with the time taken by the slowest SyntheMol-RL model.

## Random selection

We randomly selected 150 molecules as a baseline for synthesis and validation by sampling 100 molecules uniformly at random from the 31 billion REAL molecules and by sampling 50 molecules uniformly at random from the 16 billion GalaXi molecules.

## Toxicity predictions

Toxicity predictions for molecules were made using ADMET-AI version 1.2.0. ADMET-AI consists of Chemprop-RDKit models trained on 41 absorption, distribution, metabolism, excretion, and toxicity (ADMET) datasets from the Therapeutics Data Commons. We applied ADMET-AI to make predictions for all 41 properties for each molecule and then extracted the clinical toxicity (ClinTox) predictions. The clinical toxicity predictions are values in the range [0, 1] with higher values indicating a higher probability of toxicity.

## Compound synthesis

Compound synthesis was performed by Enamine as part of their Enamine REAL Space and by WuXi as part of their WuXi GalaXi. Compound purity of at least 85% was verified using liquid chromatography–mass spectrometry (LC–MS) except in cases of poor solubility, compound instability under LC–MS conditions, or non-informative LC–MS. In these cases, proton nuclear magnetic resonance (1H-NMR) was used to assess chemical purity.

## GFlowNet

GFlowNet molecules were generated using a multi-objective GFlowNet model. We specifically used the implementation of multi-objective GFlowNets from https://github.com/recursionpharma/gflownet, which implemented a multi-objective reward including binding to the target sEH, drug-likeness measure QED, molecular weight, and synthetic accessibility heuristic SAScore. We replaced this reward function with a new reward consisting of our antibacterial activity Chemprop-RDKit model ensemble, our aqueous solubility Chemprop-RDKit model ensemble, the same SAScore heuristic, and molecular weight (https://github.com/swansonk14/gflownet/tree/antibiotics). We ran the GFlowNet model with this new reward function while keeping all other parameters at their default values. After generating molecules, we selected a final set of molecules using the same filtering procedure applied to the SyntheMol compounds, but we used additional filters of SAScore ≤4 (i.e., easy synthesizability according to the SAScore heuristic) and molecular weight ≤600 to increase the likelihood of obtaining synthetically tractable compounds. Furthermore, instead of selecting the top 150 compounds by antibacterial prediction score, we only selected the top 20 since compounds had to be manually reviewed by chemists at Enamine and WuXi for synthetic accessibility.

## REINVENT 4

To generate molecules using REINVENT 4, we adapted the code at https://github.com/MolecularAI/REINVENT4 to create a multi-objective reward function using our antibacterial activity Chemprop-RDKit model ensemble, our aqueous solubility Chemprop-RDKit model ensemble, SAScore, and molecular weight (https://github.com/swansonk14/REINVENT4/tree/antibiotics). These four objectives

were balanced using the weighted geometric mean of the objectives, with a weight of 5 for antibacterial activity and a weight of 1 for the other three objectives. We experimented with weights of 1, 2, 5, 10, and 100 for antibacterial activity and found that a weight of 5 produced the most hits. The antibacterial reward was exactly the antibacterial model's prediction, the solubility reward was 1 if the solubility prediction was ≥ −4 and 0 otherwise, the SAScore reward was 1 if the SAScore was ≤4 and 0 otherwise, and the molecular weight reward was 1 if the molecular weight was ≤600 and 0 otherwise. We then used the same filtering procedure as with the GFlowNet compounds and selected the top 20 molecules by antibacterial prediction score for manual review by Enamine and WuXi.

## SyntheMol-RL ablations

We performed a series of ablation experiments to determine the effect of each of the components of SyntheMol-RL on the distribution of predicted antibacterial activity and log solubility scores across the generated molecules. We ran each ablation experiment five times with a different random seed each time. All SyntheMol-RL experiments were run with both RL-Chemprop and RL-MLP. Below are the ablation experiments:

1. Final: The final version of SyntheMol-RL. The first seed of this experiment is the one with results reported in the main text.
2. Fixed Property Weights: Instead of the dynamic property weights for antibacterial activity and aqueous solubility, fixed weights were used. The weights were (0.00 antibacterial, 1.00 solubility), (0.86 antibacterial, 0.14 solubility), (0.88 antibacterial, 0.12 solubility), (0.90 antibacterial, 0.10 solubility), (0.92 antibacterial, 0.08 solubility), (0.94 antibacterial, 0.06 solubility), (0.96 antibacterial, 0.04 solubility), and (1.00 antibacterial, 0.00 solubility). The fixed weights were chosen to balance the relative scale of the antibacterial activity predictions (ranging from 0 to 1) and the log solubility predictions (roughly ranging from -10 to 2).
3. Dynamic Temperature Target: Instead of the dynamic temperature similarity target of 0.6, similarity targets of 0.4, 0.5, 0.7, and 0.8 were used.
4. Fixed Temperature: Instead of the dynamic temperature with a similarity target, fixed temperatures of 0.01, 0.05, 0.1, 0.5, and 1.0 were used. As a point of comparison, we also ran SyntheMol-MCTS with different explore weights (the MCTS equivalent of the RL temperature parameter). The explore weights were 0.5, 1.0, 5.0, and 50.0 (the default of 10.0 was used in the final model reported in the main text).

## Antibacterial potency analyses

*S. aureus* RN4220, *S. aureus* USA300, *S. aureus* clinical isolates (VISA panel; CDC AR Isolate Bank), *Escherichia coli* BW25113, *Pseudomonas aeruginosa* PAO1, *Klebsiella pneumoniae* ATCC 43816, *Enterococcus faecium* ATCC 19434, and *Acinetobacter baumannii* ATCC 17978 were grown overnight at 37 °C in 3 ml of LB medium with shaking. Overnight cultures were then diluted 1:10,000 into fresh LB. Cells were then introduced to twofold serial dilutions of each compound under investigation in a final volume of 100 µl in Costar 96-well flat-bottom plates. Plates were incubated at 37 °C without shaking (*E. coli* BW25113, *P. aeruginosa* PAO1, *K.*

*pneumoniae* ATCC 43816, *E. faecium* ATCC 19434, and *A. baumannii* ATCC 17978) or with shaking at 900 rpm (*S. aureus* RN4220, *S. aureus* USA300, and *S. aureus* clinical isolates) until untreated control cultures reached the stationary phase. Plates were then read at 600 nm using a BioTek Synergy Neo2 plate reader.

### C.R.E.A.M. assay

*S. aureus* USA300 cells were grown overnight in 3 ml LB medium at 37 °C with shaking. ~$10^6$ CFU in 100 µl liquid LB was deposited onto solid LB agar plates. ~100 mg of control or SyntheMol-RL-generated molecule in Glaxal Base was deposited onto inoculated agar plates, and plates were incubated for 18 h at 37 °C. For chemical preparation, molecules were weighed and solubilized in 10% DMSO and then added to 100 mg Glaxal Base to a final concentration of 2% w/v. The solution was mixed thoroughly to ensure an even distribution of the compound in the Glaxal Base carrier. For the control, the same amount of vehicle (DMSO) was measured and mixed with Glaxal Base.

### Bacterial cell killing

*S. aureus* USA300 cells were grown overnight in 3 ml LB medium at 37˚C with shaking and diluted 1:10,000 into fresh LB. In 96-well flat-bottom plates, cells were grown to the required density in a final volume of 100 µl at 37˚C with shaking at 900 rpm, at which time the compound was added at the indicated concentration and cultures were incubated for the required duration. Cells were then pelleted in 96-well flat-bottom plates by centrifugation at 4000×*g* for 15 min at 4 °C and washed in ice-cold sterile PBS. After washing, cells were tenfold serially diluted in PBS and plated on solid LB agar medium.

### Mouse infection model

Mouse model experiments were conducted according to the guidelines set by the Canadian Council on Animal Care, using protocols approved by the Animal Review Ethics Board and McMaster University under Animal Use Protocol no. 22-04-10. Six- to eight-week-old female C57BL/6N mice were purchased from Charles River Laboratories (#027, QC, CAN). Animals were housed in a specific pathogen-free barrier facility under Containment Level 2 conditions and maintained on a 12 h light: 12 h dark cycle, which was temperature-controlled (21 °C) at 30–50% humidity. No animals were excluded from the analysis, and blinding was considered unnecessary.

Mice were pre-treated with 150 mg/kg (day T-4) and 100 mg/kg (day T-1) of cyclophosphamide to render mice neutropenic. On day T-0, mice were anesthetized using isoflurane and administered buprenorphine-sustained release as an analgesic at 0.6 mg/ml subcutaneously. A 2 cm$^2$ abrasion on the dorsal surface of the mouse was inflicted through tape-stripping to the basal layer of the epidermis using approximately 30-35 pieces of autoclave tape. Mice were immediately infected with ~$2.15 \times 10^7$ CFU *S. aureus* USA300 directly pipetted onto the wound bed. The infection was left to establish for 1 h before the first treatment with Glaxal Base supplemented with vehicle (10% DMSO) or synthecin (2% w/v). Mice ($n = 5$) were treated 1 h, 4 h, 8 h, 12 h, and 20 h post-infection with 150 mg Glaxal Base with synthecin (treatment) or DMSO (control). Mice were sacrificed at the experimental endpoint

(24 hpi), and wound tissue was aseptically collected, homogenized in PBS, and plated on solid LB agar medium to quantify bacterial load. For chemical preparation, synthecin was weighed and solubilized in 10% DMSO and then added to 150 mg Glaxal Base to a final concentration of 2% w/v. The solution was mixed thoroughly to ensure an even distribution of the compound in the carrier. For control groups, the same amount of vehicle (DMSO) was measured and mixed with Glaxal Base.

## Data availability

Training data (for *S. aureus* and aqueous solubility), molecular building blocks and reactions, and generated molecules are in Datasets EV1–12. All of this data, along with trained property prediction models and the molecules screened by VS-Chemprop, are available on Zenodo at https://doi.org/10.5281/zenodo.15391267 (ref. (Swanson et al, 2025)). Code for data processing and SyntheMol-RL molecule generation is available on GitHub at https://github.com/swansonk14/SyntheMol and on Zenodo at https://doi.org/10.5281/zenodo.15392449 (ref. (swansonk14 et al, 2025)). This code repository makes use of general cheminformatics functions from https://github.com/swansonk14/chemfunc as well as Chemprop model code from https://github.com/chemprop/chemprop, GFlowNet model code from https://github.com/swansonk14/gflownet/tree/antibiotics, and REINVENT 4 model code from https://github.com/swansonk14/REINVENT4/tree/antibiotics.

The source data of this paper are collected in the following database record: biostudies:S-SCDT-10_1038-S44320-026-00206-9.

## Peer review information

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

## Acknowledgements

This research was kindly supported by the Weston Family Foundation (JMS); the David Braley Centre for Antibiotic Discovery (JMS); the Canadian Institutes of Health Research (CIHR) (JMS); the Natural Sciences and Engineering Research Council of Canada (NSERC) (JMS); a generous gift from M and M Heersink (JMS); and the Chan-Zuckerberg Biohub (JZ). In addition, this research was kindly supported by the Knight-Hennessy Scholarship (KS) and the Stanford Bio-X Fellowship (KS); a CIHR Canada Graduate Scholarship CGS-D (GL and MMT); a CIHR Canada Graduate Scholarship CGS-M (DBC, SM, and AA). EDB was supported by a Tier 1 Canada Research Chair award and a Project Grant from CIHR (PJT 186281). We thank D Richman for helping to inspire the name SyntheMol as well as for early discussions of the model. Lastly, we thank the core facilities provided by McMaster University, specifically the Centre for Microbial Chemical Biology for their expertise in high-throughput screening and the Central Animal Facility for their assistance in animal housing and handling.

## Author contributions

**Kyle Swanson**: Conceptualization; Resources; Data curation; Software; Formal analysis; Funding acquisition; Validation; Investigation; Visualization; Methodology; Writing—original draft; Project administration; Writing—review and editing. **Gary Liu**: Conceptualization; Resources; Data curation; Software; Formal analysis; Funding acquisition; Validation; Investigation; Visualization; Methodology; Writing—original draft; Project administration; Writing—review and editing. **Denise B Catacutan**: Resources; Data curation; Formal analysis; Funding acquisition; Validation; Investigation; Visualization; Methodology; Writing—original draft; Project administration; Writing—review and editing. **Stewart McLellan**: Resources; Data curation; Formal analysis; Funding acquisition; Validation; Investigation; Methodology; Writing—review and editing. **Autumn Arnold**: Data curation; Formal analysis; Investigation; Methodology. **Megan M Tu**: Data curation; Formal analysis; Investigation; Methodology. **Eric D Brown**: Resources; Supervision; Funding acquisition; Project administration. **James Zou**: Resources; Software; Supervision; Funding acquisition; Investigation; Methodology; Writing—original draft; Project administration; Writing—review and editing. **Jonathan M Stokes**: Conceptualization; Resources; Supervision; Funding acquisition; Investigation; Methodology; Writing—original draft; Project administration; Writing—review and editing.

Source data underlying figure panels in this paper may have individual authorship assigned. Where available, figure panel/source data authorship is listed in the following database record: biostudies:S-SCDT-10_1038-S44320-026-00206-9.

## Disclosure and competing interests statement

KS is currently an employee of Flex Therapeutics and was previously a part-time employee of Greenstone Biosciences and a consultant at Merck & Co., Inc during this work. GL and DBC are consultants for Stoked Bio. JMS is a founder of Stoked Bio.

# Expanded View Figures

**Figure EV1. Analysis of *S. aureus* property predictor model and chemical spaces.**

(**A**) Normalized growth of *S. aureus* RN4220 in duplicate experiments used as the training data. (**B**) ROC and (**C**), precision–recall curves for each *S. aureus* activity property prediction model in an ensemble of ten, split by scaffold. Dark curves represent the average across all models. Area under the curve is indicated on the respective graphs. (**D**) Density plots showing the distribution of molecular weight and cLogP across (1) the *S. aureus* model training set, (2) a collection of known antibiotics from ChEMBL, and (3) the Enamine REAL and WuXi GalaXi chemical spaces explored by SyntheMol-RL. Not visualized are two molecules with molecular weights exceeding >3000 g/mol. (**E**) Same as (**D**), but focused on the region of the distribution containing the majority of molecules for visual clarity.

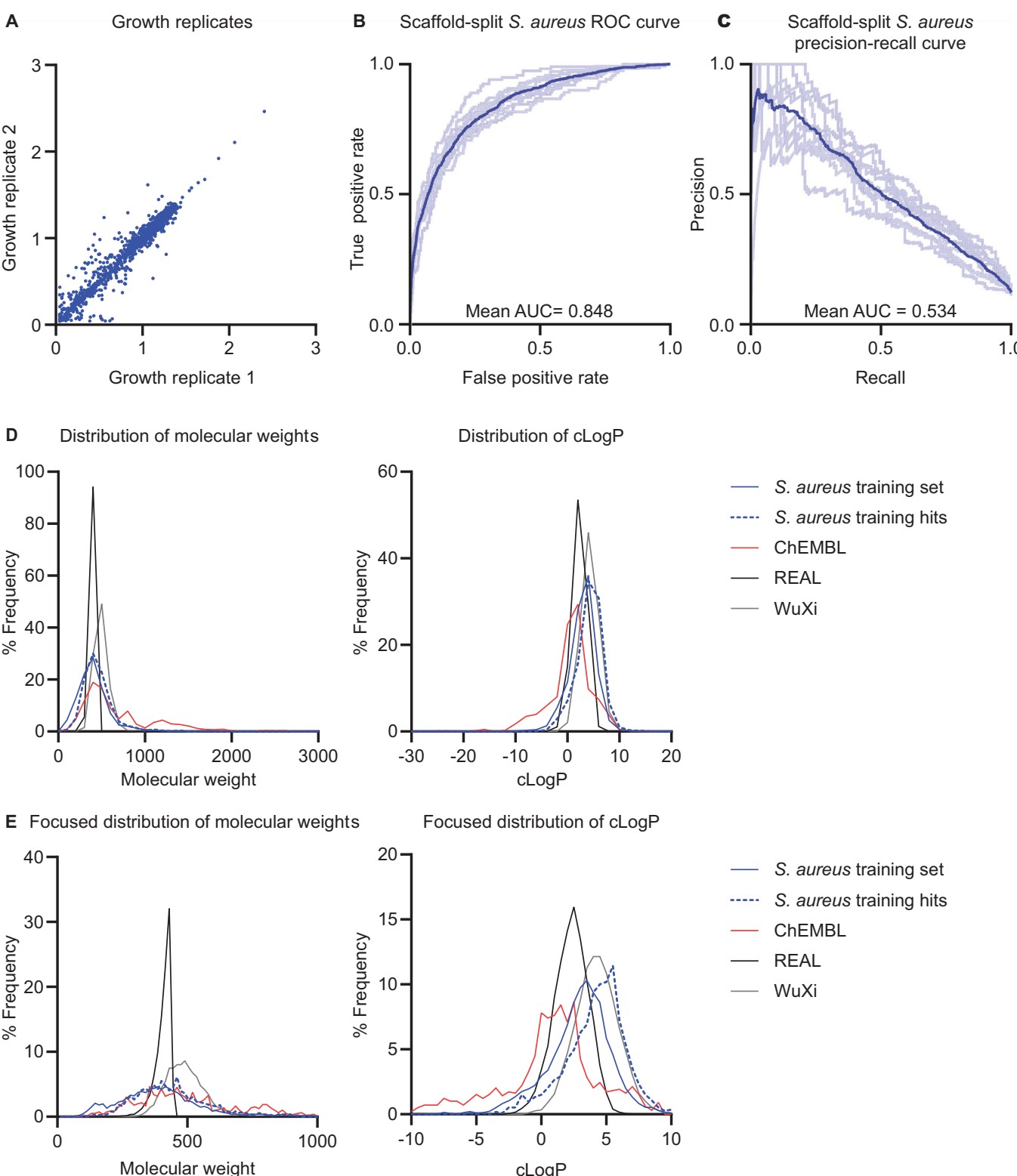

**A** Chemprop predicted *S. aureus* activity of generated compounds in all architecture combinations

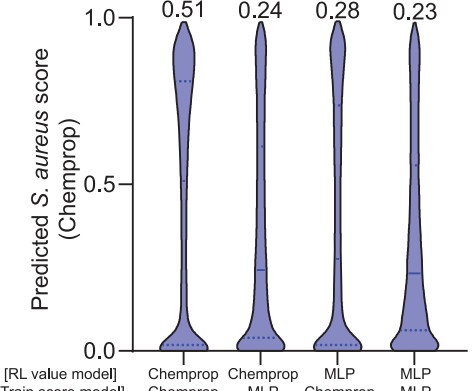

**B** MLP predicted *S. aureus* activity of generated compounds in all architecture combinations

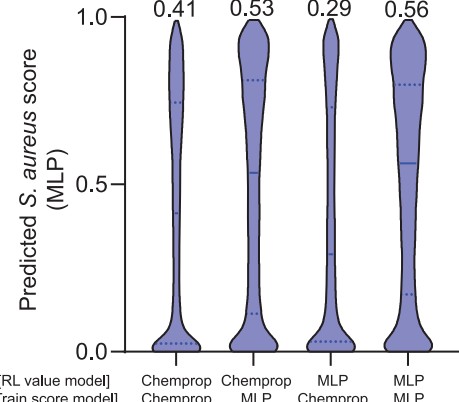

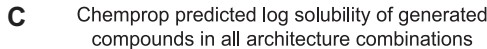

**C** Chemprop predicted log solubility of generated compounds in all architecture combinations

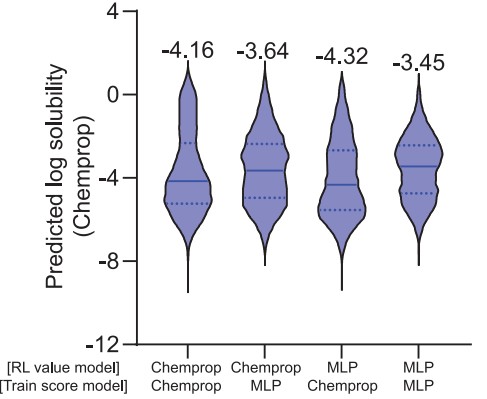

**D** MLP predicted log solubility of generated compounds in all architecture combinations

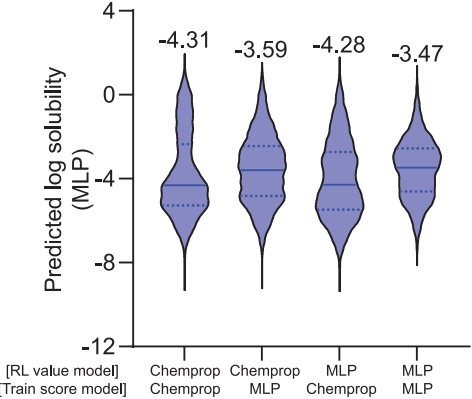

**Figure EV2.  Effects of architecture combinations on generated compound scores.**

(A) Violin plots displaying the distribution of *S. aureus* scores evaluated by a Chemprop-RDKit model after generation across all combinations of deep learning architectures acting as the RL value model and the score model used for training it during generation. (Left to right, for all panels, n = 10,983; 10,534; 9228; 11,433.) (B) Violin plots displaying the distribution of *S. aureus* scores evaluated by an MLP-RDKit model after generation across all combinations of deep learning architectures acting as the RL value model and the score model used for training it during generation. (C) Violin plots displaying the distribution of log solubility scores evaluated by a Chemprop-RDKit model after generation across all combinations of deep learning architectures acting as the RL value model and the score model used for training it during generation. (D) Violin plots displaying the distribution of log solubility scores evaluated by an MLP-RDKit model after generation across all combinations of deep learning architectures acting as the RL value model and the score model used for training it during generation.

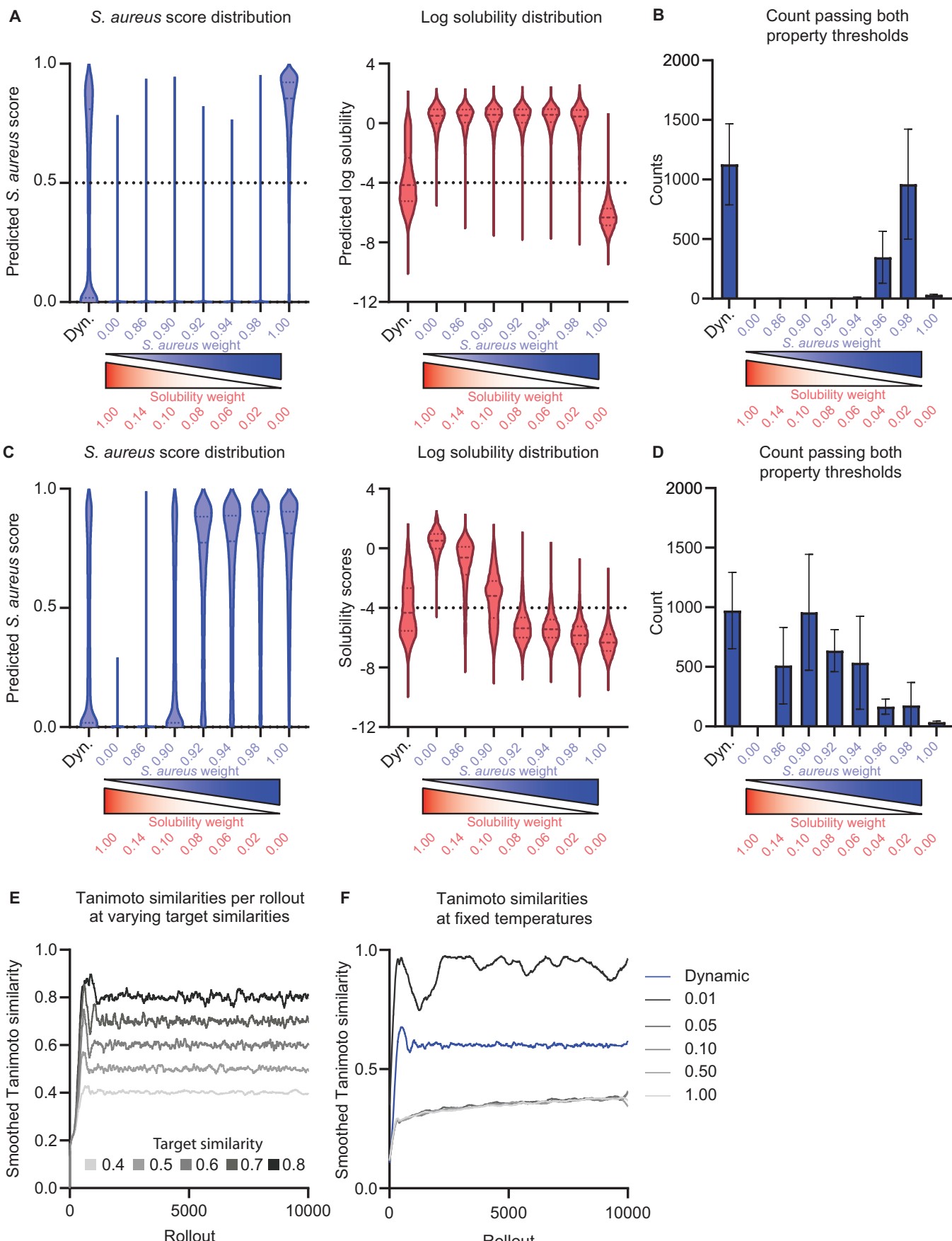

◄ **Figure EV3. SyntheMol-RL ablation experiments.**

(A) Distribution of predicted *S. aureus* activity and log solubility scores for the final dynamic weighting system (Dyn) and fixed weightings using RL-Chemprop. The sum of weights is equal to one. Left to right, for (A, C), $n = 10{,}537$; 10,282; 10,169; 10,484; 10,343; 10,613; 11,187. (B) Number of molecules generated that pass property "hit" thresholds for *S. aureus* activity and log solubility ($\geq 0.5$ and $\geq -4$, respectively). Dynamic weighting achieves optimal or near-optimal performance without need for additional fine-tuning, shown by generating the most "hit" compounds. Error bars represent the range of values found across five differently seeded runs at each weighting. (C) Distribution of predicted *S. aureus* activity and log solubility scores for the final dynamic weighting system and fixed weightings using RL-MLP. The sum of weights is equal to one. (D) Number of molecules generated that pass property "hit" thresholds for *S. aureus* activity and log solubility ($\geq 0.5$ and $\geq -4$, respectively). Error bars represent the range of values found across 5 differently seeded runs at each weighting. (E) Tanimoto similarities of each rollout compared to previous rollouts for RL-Chemprop set at desired target similarities in the range [0.4, 0.8]. The model can effectively generate compounds at a user-defined diversity. (F) Tanimoto similarities of each rollout compared to previous rollouts for RL-Chemprop at fixed temperatures and dynamic temperature for reference. Similarity is highly sensitive to changes in temperature, further necessitating the use of dynamic changes to output desired diversity.

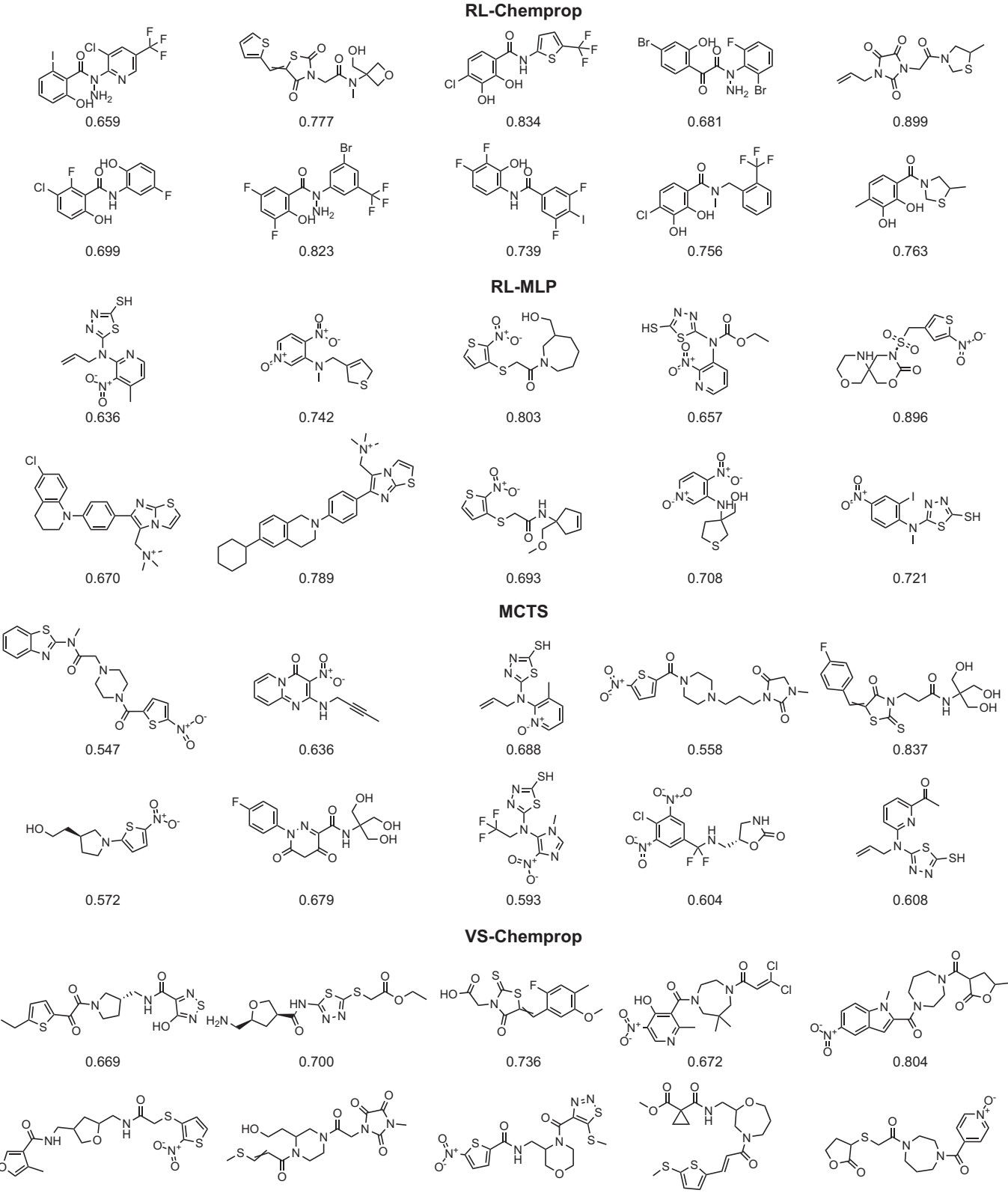

**Figure EV4. Top compounds passing our novelty and diversity filters, randomly chosen from SyntheMol and virtual screening.**

Ten random compounds from both SyntheMol-RL models, SyntheMol-MCTS, and VS-Chemprop were selected at the final in silico filtering stage "Top 150 novel diverse hits". Each compound is labeled with its respective predicted *S. aureus* score.

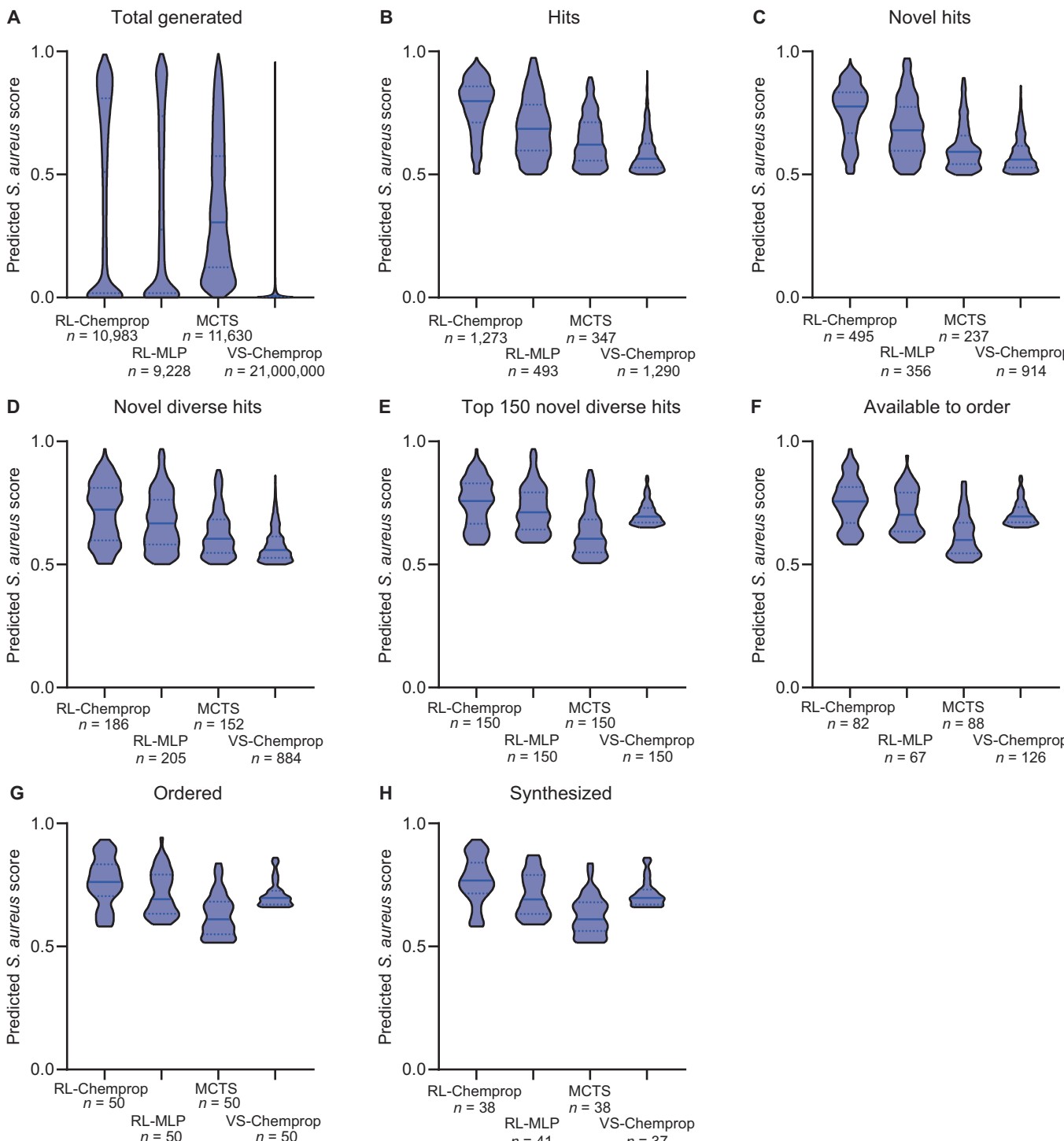

**Figure EV5.  Distribution of predicted *S. aureus* activity from SyntheMol and virtual screening at each post-hoc filtering stage.**

(**A**) All molecules generated or screened by each method. (**B**) Molecules from (**A**) with predicted *S. aureus* score ≥0.5 and predicted log solubility ≥ −4. This filter has the greatest impact on the predicted *S. aureus* score distribution, with the remaining filters having minimal impact. (**C**) Molecules from (**B**) that have a maximum Tversky similarity ≤0.6 compared to all known antibiotics in the training set and the ChEMBL antibiotics. (**D**) Molecules from (**C**) that have a maximum Tanimoto similarity to other selected molecules ≤0.6. (**E**) Molecules from (**D**) with the top 150 predicted *S. aureus* scores. (**F**) Molecules from (**E**) that are available to order from Enamine or WuXi. (**G**) Molecules from (**F**) with the lowest 50 predicted clinical toxicity values, which were ordered from Enamine or WuXi. (**H**) Molecules from (**G**) that were successfully synthesized by Enamine or WuXi and were experimentally tested.

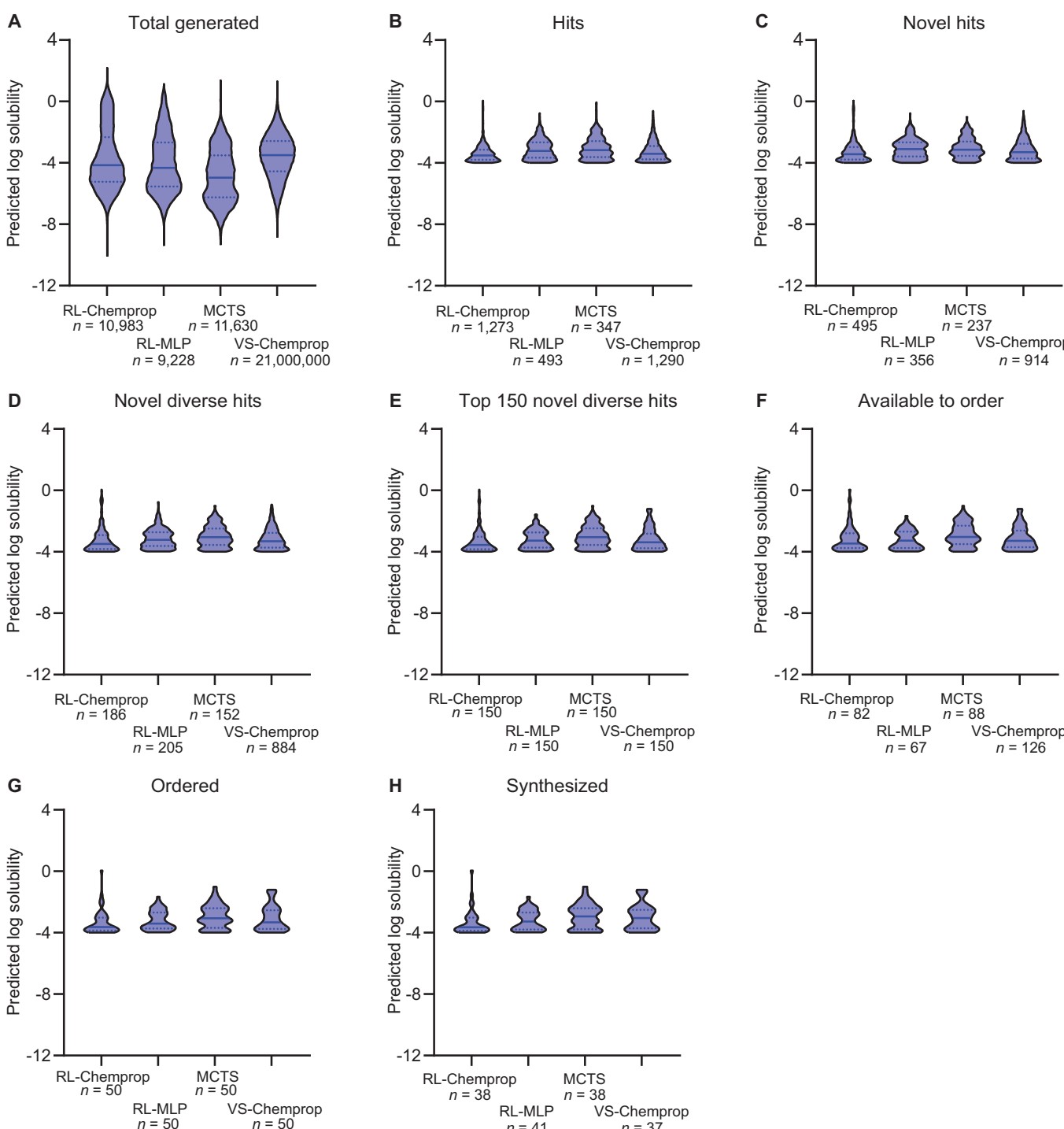

**Figure EV6. Distribution of predicted log solubility from SyntheMol and virtual screening at each post-hoc filtering stage.**

(A) All molecules generated or screened by each method. (B) Molecules from (A) with predicted *S. aureus* score ≥0.5 and predicted log solubility ≥ −4. This filter has the greatest impact on the predicted log solubility distribution, with the remaining filters having minimal impact. (C) Molecules from (B) that have a maximum Tversky similarity ≤0.6 compared to all known antibiotics in the training set and the ChEMBL antibiotics. (D) Molecules from (C) that have a maximum Tanimoto similarity to other selected molecules ≤0.6. (E) Molecules from (D) with the top 150 predicted *S. aureus* scores. (F) Molecules from (E) that are available to order from Enamine or WuXi. (G) Molecules from (F) with the lowest 50 predicted clinical toxicity values, which were ordered from Enamine or WuXi. (H) Molecules from (G) that were successfully synthesized by Enamine or WuXi and were experimentally tested.

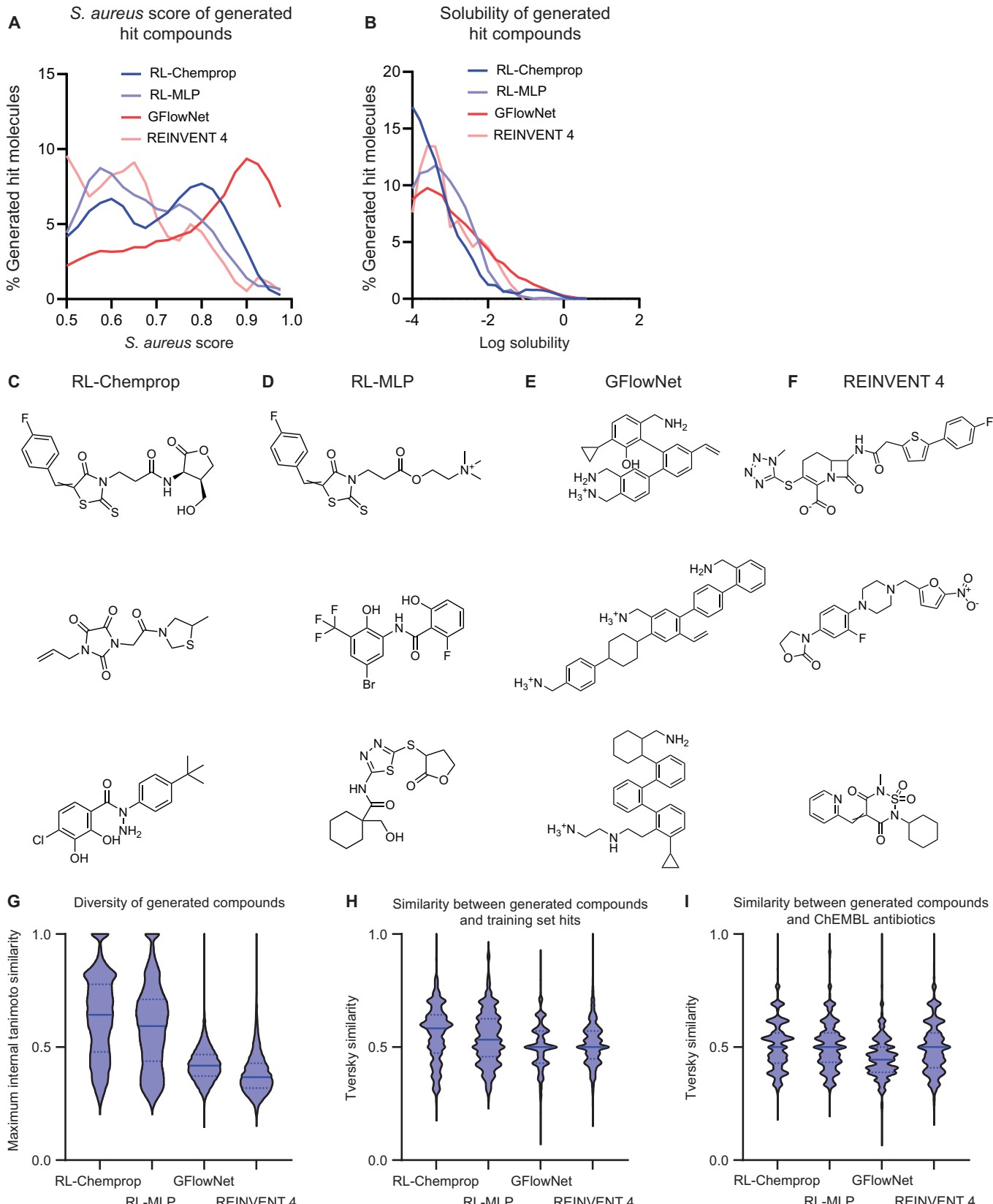

◀   **Figure EV7.   Comparisons of compounds generated by SyntheMol-RL, GFlowNet, and REINVENT 4.**

SyntheMol-RL, GFlowNet, and REINVENT 4 generated "hit" compounds that passed filters for predicted antibacterial activity (≥0.5), log solubility (≥ −4), novelty, and diversity ("Methods"). Additionally, GFlowNet and REINVENT 4 molecules were filtered for synthesizability (synthetic accessibility score SAScore ≤4) and molecular weight (weight ≤600) to ensure a fair comparison to SyntheMol-RL compounds, which are intrinsically synthetically accessible and small. RL-Chemprop generated 186 hits (total = 10,983), RL-MLP generated 205 hits (total = 9228), GFlowNet generated 1152 hits (total = 10,304), and REINVENT 4 generated 36 hits (total = 9840). Total generated compounds are further analyzed in (**G**–**I**). These hits were compared based on two properties of interest: (**A**) *S. aureus* score, and (**B**) log solubility score. Representative molecules are visualized for (**C**) RL-Chemprop, (**D**) RL-MLP, (**E**) GFlowNet, and (**F**) REINVENT 4. See Appendix Extended Discussion. (**G**) Violin plots showing the distribution of maximum Tanimoto similarities to other generated molecules in the same set for measuring structural diversity of generated compounds. (**H**) Violin plots showing the distribution of Tversky similarity calculated between generated compounds and hits from the *S. aureus* training dataset. (**I**) Violin plots showing the distribution of Tversky similarity calculated between generated compounds and the ChEMBL antibiotics.

                                     

**A**  Extended similarity evaluation of novel generated molecules versus ChEMBL dataset

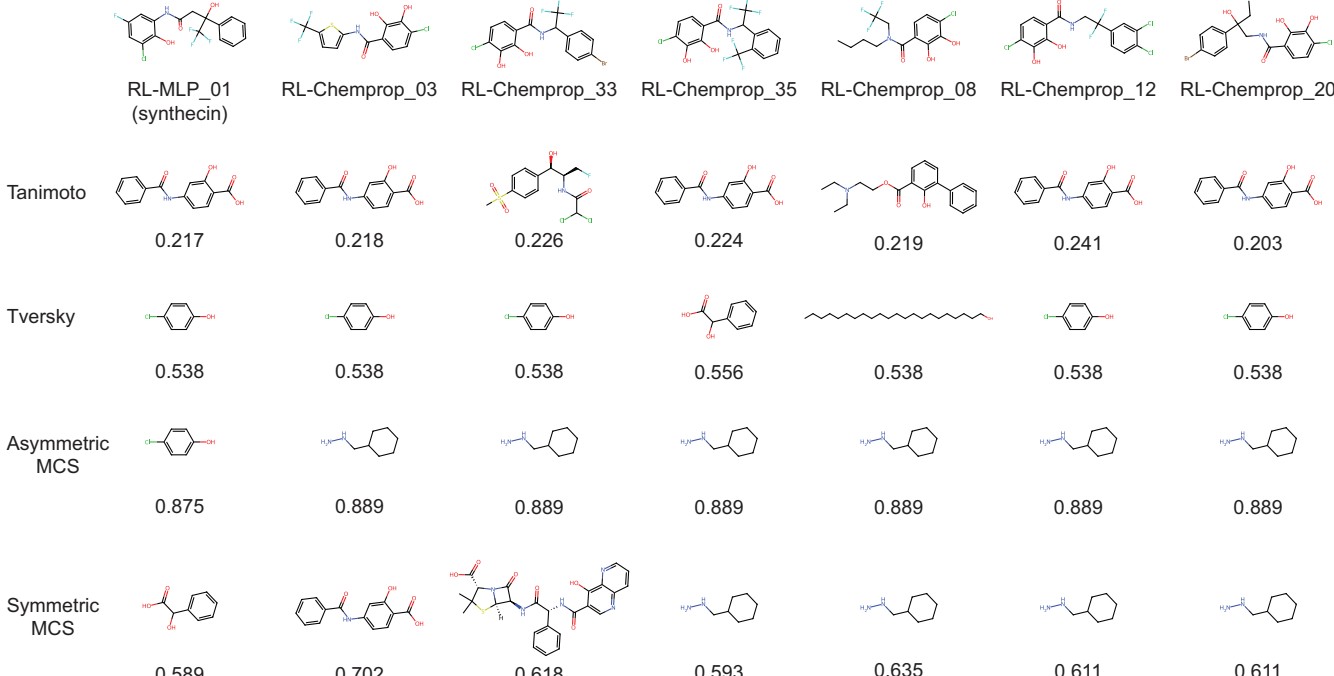

**B**  Extended similarity evaluation of novel generated molecules versus training dataset hits

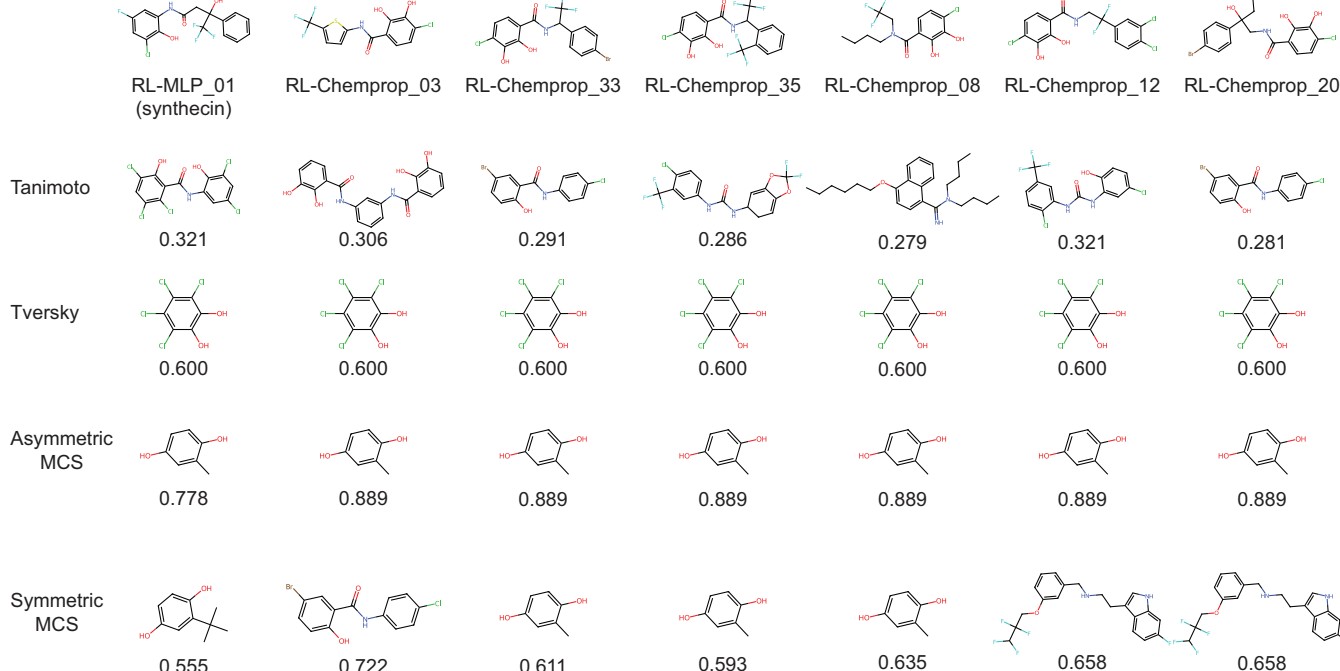

**Figure EV8. Similarity of generated compounds compared to ChEMBL antibiotics.**

(A) Molecules in the top row are highly potent, generated compounds that have passed a manual literature search for novelty using resources such as SciFinder. Each row below represents the most similar compound among the known ChEMBL antibiotics, along with the quantified similarity value. In total, four metrics of similarity were used: Tanimoto on Morgan fingerprints, Tversky on Morgan fingerprints, asymmetric Maximum Common Substructure (MCS) ratio, and symmetric Maximum Common Substructure (MCS) ratio. (B) Molecules in the top row are highly potent, generated compounds that have passed a manual literature search for novelty using resources such as SciFinder. Each row below represents the most similar hit compound in the training dataset along with the quantified similarity value. In total, four metrics of similarity were used: Tanimoto on Morgan fingerprints, Tversky on Morgan fingerprints, asymmetric Maximum Common Substructure (MCS) ratio, and symmetric Maximum Common Substructure (MCS) ratio.

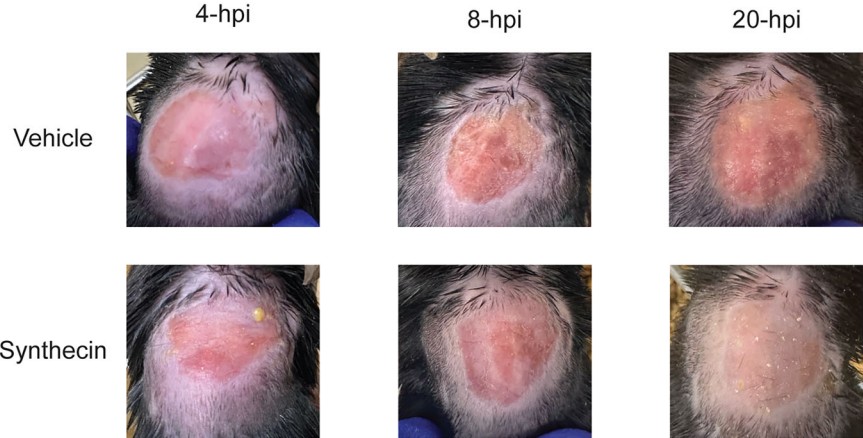

**Figure EV9.  Dorsal surface of mice with an *S. aureus* wound infection model.**

Representative images of the dorsal surface of mice after 4 h, 8 h, and 20 h of treatment with vehicle or synthecin. Note the marked inflammation on the vehicle-treated animals.

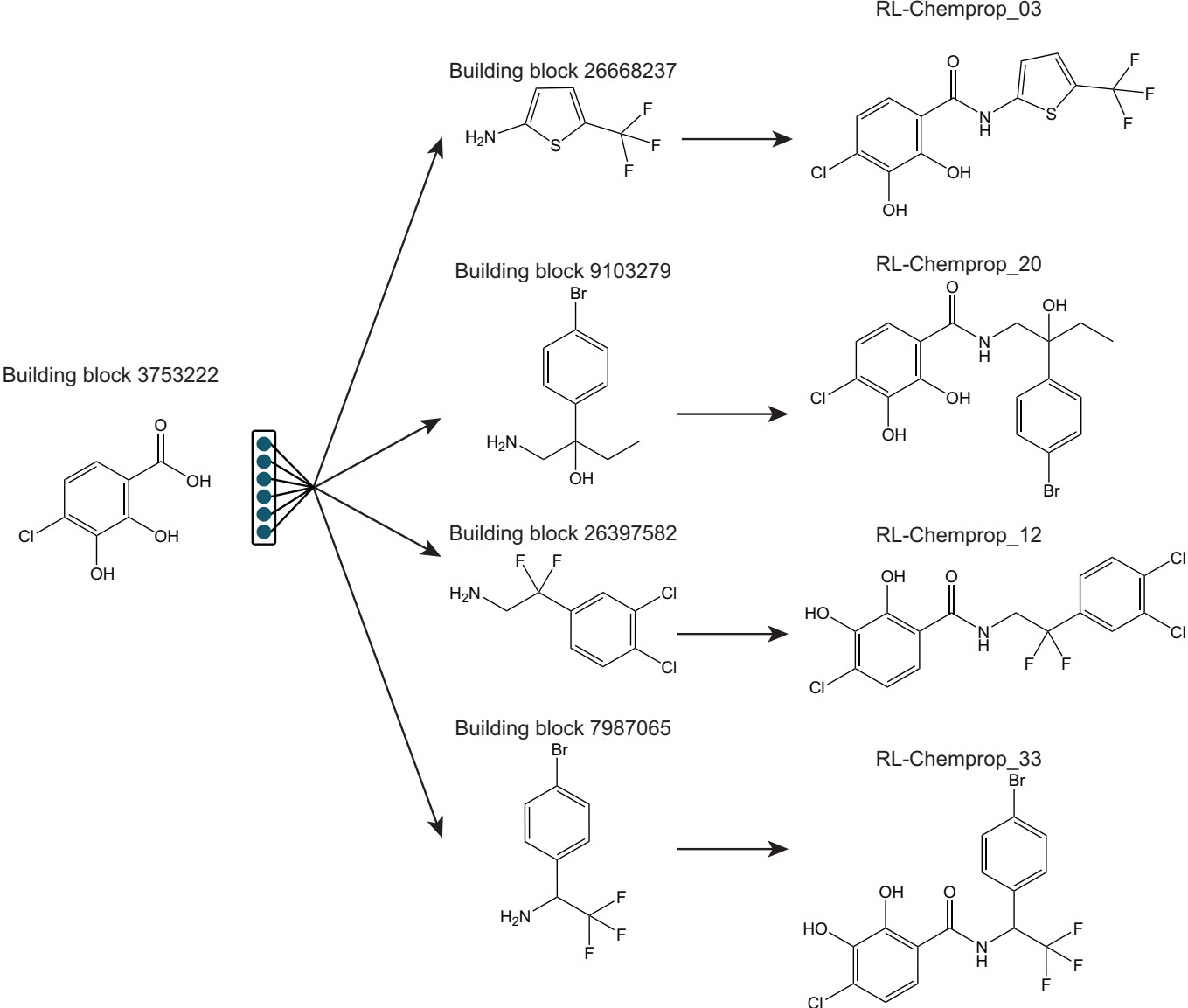

**Figure EV10. Example of molecular generations centered around particular building blocks.**

Depicted is building block 3753222, which SyntheMol-RL used multiple times when constructing hit compounds. Notably, all four compounds shown were validated to have an MIC ≤ 8 μg/ml in *S. aureus* RN4220. SyntheMol-RL seemingly employs a strategy of finding a particularly promising building block, then to satisfy the diversity objective it expands on that building block with a diverse set of "secondary" building blocks.

