## [Peer Review File · Molecular Systems Biology]

SyntheMol-RL: a flexible reinforcement learning framework for designing easily synthesizable antibiotics

Kyle Swanson, Gary Liu, Denise Catacutan, Stewart McLellan, Autumn Arnold, Megan Tu, Eric Brown, James Zou, and Jonathan Stokes

Corresponding author(s): Jonathan Stokes (stokesjm@mcmaster.ca) , James Zou (jamesz@stanford.edu)

Review Timeline:

Submission Date:	21st Feb 26
Editorial Decision:	26th Feb 26
Revision Received:	9th Mar 26
Accepted:	18th Mar 26

Editor: Jingyi Hou

Transaction Report:

This manuscript was transferred to Molecular Systems Biology following peer review at another journal.

26th Feb 2026

Manuscript Number: MSB-2026-13652-T

Title: SyntheMol-RL: a flexible reinforcement learning model for novel and synthesizable antibiotic design

Author: Kyle Swanson

Gary Liu

Denise Catacutan

Stewart McLellan

Autumn Arnold

Megan Tu

Eric Brown

James Zou

Jonathan Stokes

Dear Jon,

Thank you for transferring your manuscript to Molecular Systems Biology. We have reviewed your revised manuscript as well as the remaining concerns raised by reviewers at the previous journal. Overall, the technical issues appear to have been addressed, and we believe your study makes an interesting and important contribution to the field.

Regarding the remaining comments from the previous reviewers, please revise your manuscript to temper conclusions about novelty, both in terms of the methods and the chemical scaffolds, incorporating changes that address their suggestions. In line with Reviewer #4's recommendation, please more clearly articulate the practical utility and unique contribution of your work. Additionally, provide a point-by-point response to the previous referee reports. We will review these changes carefully. No additional analyses are required.

Please feel free to contact me if you would like to discuss any of the reviewers' comments in more detail.

On a more editorial level, please address the following issues:

1. Please remove all figures from the manuscript and retain only the figure legends at the end of the document.
2. Please resolve the discrepancy in the author name: Megan M. Tu (manuscript file) versus Megan Tu (submission system).
3. Please remove the "Authors' Contribution" section from the manuscript file.
4. Please add the missing grant number, PJT 186281, to the online submission system.
5. Code availability information should be incorporated under the "Data Availability" section, and the separate section heading should be removed.
6. Supplementary Data 1-12 should be uploaded as individual dataset files, with the legend included on a separate sheet. The nomenclature should be updated to "Dataset EV1," etc., and the corresponding callouts in the manuscript should be revised accordingly. Please delete Supplementary Data 13 and its associated callout.
7. The nomenclature and callouts for Extended Data figures (currently labeled as Figure Sx and cited as Extended Data figures) should be revised to "Figure EV1," etc.
8. The "Extended Discussion" content and all Supplementary Tables should be combined into a single PDF file titled "Appendix." The Appendix should begin with a title page that includes a Table of Contents listing each item along with its corresponding page number. The nomenclature should follow the format "Appendix Table S1," etc. All related callouts in the manuscript should be updated accordingly.
9. The references need to be formatted according to the Molecular Systems Biology reference style. Please list up to 10 co-authors of a paper before adding et al. in the reference list. Citations should be listed in alphabetical order.
10. "Competing Interests" should be renamed to "Disclosure and Competing Interests Statement"
11. Please also upload the point-by-point response from the previous rounds of peer review at the previous journal, as we require it for our records.
12. At EMBO Press we ask authors to provide source data for the main manuscript figures. We have sent you a separate email

with instructions for providing source data with your revised manuscript, including how to upload and organize the files. Additional information on source data and instruction on how to label the files are available < <https://link.springer.com/journal/44320/submission-guidelines#cms-Source-data> >. In this case, please provide source data specifically for Figure 5.

13. Please download and fill our Reagents and Tools Table template (.docx), which you can find in our author guidelines: <https://link.springer.com/journal/44320/submission-guidelines#structuredmethods>.

14. Please provide a "standfirst text" summarizing the study in one or two sentences (approximately 250 characters, including space), three to four "bullet points" highlighting the main findings and a "visual abstract" (550px width and 400-600 px height, PNG format) to highlight the paper on our homepage. Please refer to published papers for examples.

15. When you resubmit your manuscript, please download our CHECKLIST (<https://media.springernature.com/original/springer-cms/rest/v1/content/27825796/data/v1>) and include the completed form in your submission.

Please note that the Author Checklist will be published alongside the paper as part of the transparent process.

16. The manuscript sections should be in the following order: Title page - Abstract - Introduction - Results - Discussion - Methods -Data Availability - Acknowledgments - Disclosure and Competing Interests Statement - References - Figure Legends - (Main Tables with legends if applicable) - Expanded View Figure Legends.

17. Please address the following issues in figure legends:

- Please note that information related to n is missing in the legends of figures 3B, C, E, F; EV2 A-D; EDF 3 A-D; EV7 G-I
- Please note that the error bars are not defined in the legends of figures EDF3 B, D.

Use the following link to submit your revised paper:

Thank you for submitting this interesting paper to Molecular Systems Biology.

Kind regards,
Jingyi

Jingyi Hou, PhD
Senior Editor
Molecular Systems Biology

*** PLEASE NOTE *** As part of the EMBO Press transparent editorial process initiative (see our Editorial at <https://dx.doi.org/10.1038/msb.2010.72> , Molecular Systems Biology will publish online a Review Process File to accompany accepted manuscripts. When preparing your letter of response, please be aware that in the event of acceptance, your cover letter/point-by-point document will be included as part of this File, which will be available to the scientific community. More information about this initiative is available in our Instructions to Authors. If you have any questions about this initiative, please contact the editorial office (msb@embo.org).

Reviewer #1:

I appreciate the authors' detailed response. However, my fundamental concerns regarding the novelty of the chemical outputs and the methodological advance remain unresolved. The authors confirm that Synthecin shares an identical core scaffold with known patented compounds, differing only by a single halogen substitution (F vs. Cl). Medicinally, this constitutes a ~99% similar analog, not a structurally novel entity. Beyond Synthecin, it is notable that most active compounds presented in the manuscript (Fig. 4b) share this same structural motif. From a medicinal chemistry perspective, they possess the same parent structure and are derivatives of one another, rather than representing distinct, novel scaffolds. This pattern is puzzling and further weakens the claim of discovering novel chemical matter. The authors' defense—that the antibacterial activity is "novel to the model" and represents a "new use"—misinterprets the core of the critique. The issue is not the validity of drug repurposing per se, but its inadequacy as evidence for the model's purported ability to discover novel chemical matter. Featuring a known scaffold as the flagship output fundamentally contradicts the central claim of exploring a "46 billion" synthesizable space to find new scaffolds. A rigorous prior-art check, a standard and essential step in translational science, would have identified this issue and prevented the presentation of this compound as a demonstration of scaffold novelty.

Thank you for sharing your concerns. We agree that the compounds generated by SyntheMol are similar to compounds that exist in the patent literature, but we would like to reiterate that the patent compounds were identified with insecticidal/acaricidal activity, which is not relevant for antibacterial activity. Indeed, we do not claim that SyntheMol-RL generates molecules that have never been synthesized or tested in any capacity before; we simply show that for a given drug discovery task like antibiotic design, SyntheMol-RL can generate active compounds that are dissimilar from known compounds *for that particular application* (in our case, dissimilar from known antibiotic compounds). This is noted in lines 393-395 of the revised manuscript, where we state that our "literature search was conducted to ensure structural novelty from molecules with reported antibacterial activity from literature," not to ensure novelty with respect to any molecule in the literature. To further emphasize this point, we have now changed the abstract to say that "seven [compounds] passed our structural novelty filters that compared them to known antibiotics" (lines 45-46).

Regarding the analysis that the generated compounds are "novel to the model," this is the standard analysis for generative AI models for molecule design. For example, [1-3] claim that their AI models generate "novel" molecules by calculating the number of SMILES that do not appear exactly in the training set, even if they only differ by one atom. By comparing generated molecules to the training set, these works are measuring novelty with respect to what the model was trained on. We consider this a relatively weak form of novelty, which is why we use a much stricter definition of novelty that requires a maximum Tversky similarity of 0.6 to any training set molecule. Additionally, we evaluate the Tanimoto similarity and two versions of the maximum common substructure similarity between our hits and the training set to further illustrate novelty. This analysis goes beyond many novelty analyses in the literature, and it shows that SyntheMol-RL can generate active molecules that differ significantly from the molecules it was trained on, thus proving the generalizability and power of the model. To further verify the practical utility of SyntheMol-RL to generate molecules that are unlike any existing antibiotics, even beyond those in the training set, we also use the same similarity metrics to ensure novelty with respect to all known antibiotics in ChEMBL. For even more certainty, we perform a careful literature search to check for similarity to other known antibiotics. The generated molecules will of course be "known" scaffolds in the sense that they are in a commercially available chemical space and

may have been used for other applications, but their newly discovered antibacterial activity is what constitutes a novel discovery in the antibiotics field.

Due to the similarity between some of our seven hit compounds and salicylanilides, which are known to have antibiotic activity, we have softened our claims of novelty. Please see our response to Reviewer #4 below for a complete list of the changes to the text regarding novelty.

[1] Moret et al. Generative molecular design in low data regimes. *Nature Machine Intelligence* **2**, 171–180 (2020).

[2] Kotsias et al. Direct steering of de novo molecular generation with descriptor conditional recurrent neural networks. *Nature Machine Intelligence* **2**, 254–265 (2020).

[3] Born and Manica. Regression Transformer enables concurrent sequence regression and generation for molecular language modelling. *Nature Machine Intelligence* **5**, 432–444 (2023).

Regarding methodological novelty, the authors' analogy comparing the MCTS-to-RL transition to the RNN-to-Transformer shift is overstated. The described technical changes (e.g., a deep value network, policy temperature adjustment) represent incremental engineering improvements within the well-established paradigm of reaction-rule-based RL assembly. This does not constitute the paradigm shift suggested by the analogy. As also noted by Reviewer #3 and supported by prior literature, the core conceptual framework remains closely aligned with existing works.

We agree that our MCTS to RL change is not as fundamental as the RNN to Transformer change, and we only intended the analogy to show that ML models that operate on the same underlying data structure can still have dramatic differences in their model architecture. Thus, the reuse of a tree structure for representing molecular building blocks and reactions does not necessarily undermine the technical novelty of our RL model. Despite similarities to the literature, we still believe that SyntheMol-RL represents an important technical advance because it is not trivial to adapt any RL-based molecule generation model to generate truly synthesizable molecules. Indeed, this can readily be inferred from the majority of prior works never synthesizing any of their *in silico* generated molecules (lines 539-545).

Ultimately, the manuscript's narrative is built on a dual claim: (A) a novel method for exploring vast synthesizable spaces, and (B) the discovery of novel antibacterial candidates. The selection of a non-novel scaffold as the primary experimental validation severely undermines claim (B), which in turn critically weakens the support for claim (A). There is a fundamental inconsistency between the proposed methodological advancement and the chemical evidence presented.

Thank you for this critique. We respectfully maintain that claim (A) holds because SyntheMol-RL does indeed explore vast synthesizable chemical spaces for promising compounds. On claim (B), we refer the Reviewer to our response about novelty above.

In conclusion, while the revisions demonstrate effort, they primarily address peripheral aspects rather than this core scientific inconsistency. The work applies a refined version of an existing paradigm to a practical problem—yielding a useful but derivative outcome that does not meet the conceptual advancement expected.

We appreciate your perspective, but we politely disagree. We believe that SyntheMol-RL is an important contribution in that our RL method enables the generation of molecules that can be easily synthesized and tested *in vitro* and *in vivo*, with promising therapeutic potential.

Reviewer #3:

Re Technical Novelty

Thank you for specifying the differences between this model and the original SyntheMol work. We agree that there are differences in the model, which make it an improvement. Regarding technical novelty, I maintain that there is no core machine learning advance, but this is a thoughtful application. The ability to generate synthesizable molecules at scale, optimizing across several tasks, is useful.

Thank you for recognizing the improvements in SyntheMol-RL and noting the usefulness of synthesizable molecule design with multi-parameter optimization. We believe that the improvements introduced by our RL method, along with multi-parameter optimization applied to generating synthesizable molecules from a multi-billion molecule space, constitute an important machine learning advance. This is because most previous works on generative AI for molecule design did not design methods that can generate easily synthesizable compounds, which requires significant technical changes to the model architecture to exclusively and efficiently generate molecules using building blocks and chemical reactions from commercially available combinatorial chemical spaces. This may be a reason that most prior works did not physically synthesize and test any of their *in silico* generated molecules (lines 539-545). Contrarily, due to the machine learning advances introduced by SyntheMol-RL, we could synthesize and test 79 molecules generated by our model, which unlocks the practical utility of generative AI for small molecule drug design.

Re Limitations of WuXi and Enamine Spaces and Building Block Methods

Thank you for expanding the discussion of the limitations of the WuXi and Enamine spaces. I thank you for adding discussion of how users could expand the chemical space. I am convinced of your arguments that at the scale of thousands of molecules, manual review is not plausible, so having molecules that are purchasable by nature is useful. However, I would appreciate clarification in the manuscript that this is searching a miniscule fraction of the chemical space. Some readers may think that the space explored by SyntheMol-RL is covering much of the drug-like chemical space. Perhaps around the discussion on Synthemol-RL to GFlowNet and REINVENT, I would appreciate that a line of commentary that this is exploring a larger, more unconstrained space compared to SyntheMol-RL.

Thank you for recognizing our expanded discussion of the limitations of these chemical spaces. The less constrained nature of GFlowNet and REINVENT 4 is noted in lines 364-367: "A final computational evaluation was conducted prior to *in vitro* testing, which benchmarked SyntheMol-RL against GFlowNet and REINVENT 4, two state-of-the-art-generative models for drug design that are not constrained to the Enamine and WuXi chemical spaces."

Re Molecular Novelty

You state that: "The fact that only a subset of these compounds was effective *in vitro*, despite all of them having high activity scores according to the property predictors, is a limitation of the property predictors rather than a limitation of SyntheMol-RL"

Because the manuscript positions SyntheMol-RL as an end-to-end system intended to be useful to chemists and biologists (the abstract describes it as "a powerful tool for drug design across therapeutic domains"), I view limitations of the property predictors as limitations of the overall system. This argument would be more compelling in the context of a paper focused on fundamental reinforcement learning advances. Here, however, the applicability claims are weakened if a core component of the pipeline (e.g., Chemprop) underperforms relative to

available alternatives. With this being said, I do agree that this does not reflect poorly on their reinforcement learning system and how they focused on generating synthesizable molecules.

Thank you for recognizing that the limitations of the property predictors are distinct from the RL system within SyntheMol-RL. We would like to further clarify why this limitation is not fundamental to SyntheMol-RL. While the Chemprop models that were trained on antibacterial activity had inaccuracies that resulted in limited efficacy of some of our generated molecules, this is not strictly a limitation of Chemprop, but also largely due to the limited size of truly high-quality antibiotic training datasets that can be acquired. Unfortunately, there simply aren't significantly better predictors of antibiotic activity, and so it is not the case that Chemprop "underperforms relative to available alternatives" but more so that there are no better alternatives for antibacterial activity prediction. With a much larger dataset and/or a different property of interest, Chemprop may have sufficient accuracy that it will no longer be the limiting factor. Notably, SyntheMol-RL is not limited to using Chemprop as the property predictor and can, in fact, use any model or function to compute the properties that serve as a reward for the RL model. Therefore, if a user has access to an accurate property prediction model, then SyntheMol-RL will no longer be limited by that property predictor. Therefore, while the property predictor was a limiting factor in our antibiotic application, that does not make it a general limiting factor of SyntheMol-RL.

Thank you for providing Extended Data Table 4, which shares 10 structures from each of the methods. I maintain that there are systematic similarities between molecules. Even in this set, from the RL-Chemprop method, 7/10 have an amide group directly connected to a benzene ring, and 5 of those are connected to another aromatic functionalized by halides. While this pattern may be influenced by modifiable filters (lines 532–536), as presented, I am not convinced that the flagship Chemprop-RL method produces sufficiently novel or diverse compounds.

Thank you for examining our additional data. While there are similarities in the generated compounds and we acknowledge that there could be greater novelty and diversity, we still believe that these compounds represent sufficiently novel and diverse compounds since they pass Tversky and Tanimoto similarity filters, which are commonly used in the literature. In fact, many comparable papers in the literature (e.g., [1-3]) compute novelty using an exact SMILES match to the training set, which is a much weaker form of novelty than the Tversky similarity filters we applied not only to the training set but also to the set of all known antibiotic compounds in ChEMBL. Even so, due to the similarities you noted, we have now softened our claims of novelty in the manuscript. Please see our response to Reviewer #4 below for a complete list of the changes in the revised text regarding novelty.

[1] Moret et al. Generative molecular design in low data regimes. *Nature Machine Intelligence* **2**, 171–180 (2020).

[2] Kotsias et al. Direct steering of de novo molecular generation with descriptor conditional recurrent neural networks. *Nature Machine Intelligence* **2**, 254–265 (2020).

[3] Born and Manica. Regression Transformer enables concurrent sequence regression and generation for molecular language modelling. *Nature Machine Intelligence* **5**, 432–444 (2023).

Finally, I am not convinced that many of the molecules tested and presented in Figure 4 are substantially different enough from salicylanilides. As stated in my past review, RL_Chemprop33 and 35 have a core that's just different by one carbon spacer. The authors rebut saying that "Salicylanilide molecules are fairly rigid, defined by a hydroxybenzamide core with strong intramolecular hydrogen bonding between the phenolic hydroxyl and amide carbonyl groups", attempting to draw a contrast between their molecules and salicylanilides. However, it

is crucial to note that RL-Chemprop_20, RL-Chemprop_33, RL-Chemprop_35, RL-Chemprop_03, and RL-Chemprop_12 have the exact same distance between the amide group and the phenolic hydroxyl. Thus, these molecules would all participate in the intramolecular hydrogen bonding that the authors claim is characteristic of salicylanilides. The spacers are on the other side of the amide linkage. Thus, the analysis that the authors share on this point does not accurately represent the differences between many of their molecules and salicylanilides. I do not believe that their claim of only a few compounds sharing “minor similarity with salicylanilides” (Line 386) is justified. I also do not believe that these molecules in particular can be characterized as “structurally novel after detailed literature searches” (line 32).

Thank you for your analysis of the similarities between salicylanilides and our generated compounds. We have modified the text to better acknowledge these similarities. On lines 409-410, we changed “a few of these compounds shared some minor similarity with salicylanilides” to “a few of these compounds shared similarity with salicylanilides.” The rest of the discussion in that paragraph (lines 410-425) only comments on the dissimilarities between synthecin and salicylanilides and does not extend that analysis to those other compounds. Please see our response to Reviewer #4 below for a complete list of the changes in the text regarding novelty.

The authors raise the point correctly that synthecin is more different from this core, but in my previous review I had not highlighted the similarity of synthecin in this section. From my previous review: “However, this structure is quite similar to many of the molecules the authors present. RL_Chemprop_33, RL_Chemprop_35 differ from the salicylanilide by a single carbon spacer, and RL_Chemprop_10, RL-Chemprop_12 differ by two carbon spacers”

We acknowledge that those compounds are more similar to salicylanilides, and we appreciate your agreement that synthecin is more different.

It is true that synthecin would not participate in this proposed interaction, and to the credit of the authors, it is more different from salicylanilides than the other compounds the authors proposed. I agree with the authors that it is sufficiently different that it is not simply a salicylanilide derivative. However, I maintain it does share similar structural elements as the rest of the molecules, despite the reversed orientation of the amide group relative to the phenolic hydroxyl group.

Thank you for acknowledging the dissimilarity of synthecin. We agree that it contains some shared structural elements but highlight the notable differences that make it a potent and translationally promising hit generated by SyntheMol-RL.

Reviewer #4:

The revision has addressed all of the comments I previously raised.

We're pleased that we were able to address your comments!

With respect to the questions about computational novelty raised by other reviewers, I understand the concern that the method does not necessarily generate entirely new chemical scaffolds. However, I see the main contribution of this work as demonstrating a practical strategy for discovering synthesizable molecules with targeted properties, rather than focusing exclusively on unprecedented structural novelty. If this distinction is clearly articulated in the manuscript, I believe the contribution is well framed. Even if the method's impact is confined to

enabling computational drug repurposing rather than new scaffold discovery, such practical utility itself represents a meaningful contribution.

Thank you for appreciating our contribution to discovering easily synthesizable molecules with desired properties. To soften our claims of structural novelty, we have revised the title by removing the words “novel and” (lines 2-3).

We changed the abstract from “seven [compounds] were structurally novel after detailed literature searches” to “seven [compounds] passed our structural novelty filters that compared them to known antibiotics” (lines 45-46). In the abstract, we also removed “novel and” in the sentence “These results validate SyntheMol-RL’s ability to generate synthetically accessible candidate antibiotics” (lines 47-48).

In the introduction, we modified the phrase “finding more compounds that are novel, diverse, and predicted to be antibacterial” to “finding more compounds that pass our novelty and diversity filters and are predicted to be antibacterial” (lines 126-127). We also removed the word “novel” in the sentence “These results demonstrate that SyntheMol-RL is an effective and flexible framework for *novel* drug design applications” (lines 133-134).

In the main text, we changed “All molecules that failed the detailed manual novelty check were removed from further investigations, leaving six *novel* compounds from RL-Chemprop, one *novel* compound from RL-MLP, and no *novel* compounds from VS-Chemprop and MCTS” to remove the italicized appearances of the word “novel” (lines 407-409). In the sentence, “This provides strong evidence that all seven molecules are able to overcome a wide array of prevalent antibiotic resistance determinants, *likely owing to their structural novelty relative to clinical antibiotics*” we removed the italicized phrase (lines 449-451). In the sentence “To address this urgent and growing unmet need, we designed *novel* antibacterial molecules that display promising activity in such resistant *S. aureus* strains” we removed the italicized word “novel” (lines 575-576). In the sentence “We identified seven *structurally novel* compounds that retained potent activity against *S. aureus* USA300” we removed the italicized words “structurally novel” (lines 581-582). In the sentence “While SyntheMol-RL succeeded in designing a diverse set of *novel* compounds with high property prediction scores” we removed the italicized word “novel” and added the phrase “that passed our novelty filters” (lines 597-598).

In the Figure 4 legend in the sentence “Growth inhibition of *S. aureus* RN4220 by the seven *structurally novel* molecules that passed the manual literature search” we removed the italicized words “structurally novel” and changed the end to “manual literature search for novelty” (lines 1404-1405). In the sentence “Heat maps depicting the MICs of the seven *novel* compounds” we removed the italicized word “novel” (lines 1408-1409).

In the Figure EV4 (previously Extended Data Figure 4) legend, we changed “Top novel and diverse compounds randomly chosen from SyntheMol and virtual screening” to “Top compounds passing our novelty and diversity filters randomly chosen from SyntheMol and virtual screening” (lines 1488-1489). We also removed the sentence “Each method leads to structurally novel and diverse hits” (lines 1491-1492).

In the Figure EV8 (previously Extended Data Figure 8) legend, in the sentence “Similarity of *novel* generated compounds compared to ChEMBL antibiotics” we removed the italicized word “novel” (line 1536).

Regarding the novelty of the in vitro tested candidates, I think the authors' newly added statement is helpful: "While SyntheMol-RL succeeded in designing a diverse set of novel compounds with high property prediction scores, only a subset of the generated compounds showed in vitro activity. This demonstrates the need to improve the accuracy of property prediction models so that high scoring compounds generated by SyntheMol-RL translate into real hits in the laboratory." This clarification appropriately acknowledges the current limitations of the property predictor. The predictor may both overestimate and underestimate activity. In particular, it may underestimate the activity of more structurally novel compounds, leading to their exclusion during filtering steps. The structural similarity observed within the final candidate set does not necessarily undermine the approach. To some extent, it reflects the algorithm's ability to learn functional groups associated with the desired properties.

We are pleased that our addition was helpful. We agree with your analysis that the predictor's underestimates may contribute to reduced novelty and/or diversity.

Overall, I find that the manuscript has been substantially improved, and the key limitations are now discussed.

Thank you for your valuable comments.

Editorial comments:

On a more editorial level, please address the following issues:

1. Please remove all figures from the manuscript and retain only the figure legends at the end of the document.

All main figures and supplementary figures are now uploaded as individual files and removed from the main manuscript.

2. Please resolve the discrepancy in the author name: Megan M. Tu (manuscript file) versus Megan Tu (submission system).

This has been fixed to 'Megan M. Tu' in the submission system.

3. Please remove the "Authors' Contribution" section from the manuscript file.

Author contributions in the revised manuscript have been removed. Author contributions have now been added to the submission system.

4. Please add the missing grant number, PJT 186281, to the online submission system.

A new funding source has been added to the online submission system, 'Canadian Institutes of Health Research | PJT 186281'.

5. Code availability information should be incorporated under the "Data Availability" section, and the separate section heading should be removed.

Code availability has now been merged in the Data Availability section.

6. Supplementary Data 1-12 should be uploaded as individual dataset files, with the legend included on a separate sheet. The nomenclature should be updated to "Dataset EV1," etc., and the corresponding callouts in the manuscript should be revised accordingly. Please delete Supplementary Data 13 and its associated callout.

All Supplementary Data files are now labeled as Dataset EV files. Additionally, all callouts in the manuscript text have been updated accordingly. Dataset 13 and its text callouts have been removed from the manuscript.

7. The nomenclature and callouts for Extended Data figures (currently labeled as Figure Sx and cited as Extended Data figures) should be revised to "Figure EV1," etc.

Callouts to Extended Data figures have been revised from Extended Data Fig. X to Figure EVX.

8. The "Extended Discussion" content and all Supplementary Tables should be combined into a single PDF file titled "Appendix." The Appendix should begin with a title page that includes a Table of Contents listing each item along with its corresponding page number. The nomenclature should follow the format "Appendix Table S1," etc. All related callouts in the manuscript should be updated accordingly.

Extended Discussion and Supplementary Tables are now found within a new Appendix file, which includes a table of contents. Accordingly, callouts in the text have changed from 'Supplementary Table X' and 'Extended Discussion' to 'Appendix Table SX' and 'Appendix Extended Discussion' respectively.

9. The references need to be formatted according to the Molecular Systems Biology reference style. Please list up to 10 co-authors of a paper before adding et al. in the reference list. Citations should be listed in alphabetical order.

All in-text citations, as well as the reference list, have been updated to Molecular Systems Biology's reference style.

10. "Competing Interests" should be renamed to "Disclosure and Competing Interests Statement"

This header has been renamed accordingly.

11. Please also upload the point-by-point response from the previous rounds of peer review at the previous journal, as we require it for our records.

We have now responded to all comments from the previous round of review and uploaded them as this reviewer and editor response document.

12. At EMBO Press we ask authors to provide source data for the main manuscript figures. We have sent you a separate email with instructions for providing source data with your revised manuscript, including how to upload and organize the files.

Additional information on source data and instruction on how to label the files are available < <https://link.springer.com/journal/44320/submission-guidelines#cms-Source-data> >.

In this case, please provide source data specifically for Figure 5.

A new folder has been added, entitled 'Figure 5_source_data', which contains all relevant source data for Figure 5. We have also completed and uploaded the source data checklist as 'MSB-2026-13652-T_SourceDataChecklist.xlsx'.

13. Please download and fill our Reagents and Tools Table template (.docx), which you can find in our author guidelines: <https://link.springer.com/journal/44320/submission-guidelines#structuredmethods>.

A completed version of the table has been uploaded as 'Reagent and Tools Table.docx'.

14. Please provide a "standfirst text" summarizing the study in one or two sentences (approximately 250 characters, including space), three to four "bullet points" highlighting the main findings and a "visual abstract" (550px width and 400-600 px height, PNG format) to highlight the paper on our homepage. Please refer to published papers for examples.

A new header, 'Standfirst Text' can be found in the revised manuscript file prior to the abstract. This section contains a brief summary, three bullet points highlighting the importance of this study, and a visual abstract. The visual abstract has also been supplied as its own file, 'Figure_abstract.png'

15. When you resubmit your manuscript, please download our CHECKLIST (<https://media.springernature.com/original/springer-cms/rest/v1/content/27825796/data/v1>) and include the completed form in your submission. *Please note* that the Author Checklist will be published alongside the paper as part of the transparent process.

A completed version of this table is uploaded as 'Author Checklist.xlsx'.

16. The manuscript sections should be in the following order: Title page - Abstract - Introduction - Results - Discussion - Methods -Data Availability - Acknowledgments - Disclosure and Competing Interests Statement - References - Figure Legends - (Main Tables with legends if applicable) - Expanded View Figure Legends.

The manuscript has been edited to follow the recommended order. A section has now been added for the Main Table.

17. Please address the following issues in figure legends:

- Please note that information related to n is missing in the legends of figures 3B, C, E, F; EV2 A-D; EDF 3 A-D; EV7 G-I
- Please note that the error bars are not defined in the legends of figures EDF3 B, D.

All figures now have n defined in either figure legends or embedded within the figure. New descriptions of error bars have now been added for Figure EV3B,D.

18th Mar 2026

Manuscript number: MSB-2026-13652R

Title: SyntheMol-RL: a flexible reinforcement learning model for novel and synthesizable antibiotic design

Dear Jon,

Thank you again for sending us your revised manuscript. We are now satisfied with the modifications made and I am pleased to inform you that your paper has been accepted for publication.

You may qualify for financial assistance for your publication charges - either via a Springer Nature fully open access agreement or an EMBO initiative. Check your eligibility: <https://link.springer.com/journal/44320/how-to-publish-with-us>

Kind regards,
Jingyi

Jingyi Hou, PhD
Senior Editor
Molecular Systems Biology

>>> Please note that it is Molecular Systems Biology policy for the transcript of the editorial process (containing referee reports and your response letter) to be published as an online supplement to each paper. If you do NOT want this, you will need to inform the Editorial Office via email immediately. More information is available here: <https://link.springer.com/partners/embo-press/editorial-policies#Peer%20review>